# Myosin 7 and its adaptors link cadherins to actin

I-Mei Yu[1,*], Vicente J. Planelles-Herrero[1,2,*], Yannick Sourigues[1], Dihia Moussaoui[1], Helena Sirkia[1], Carlos Kikuti[1], David Stroebel[3], Margaret A. Titus[4,**] & Anne Houdusse[1,**]

Cadherin linkages between adjacent stereocilia and microvilli are essential for mechanotransduction and maintaining their organization. They are anchored to actin through interaction of their cytoplasmic domains with related tripartite complexes consisting of a class VII myosin and adaptor proteins: Myo7a/SANS/Harmonin in stereocilia and Myo7b/ANKS4B/Harmonin in microvilli. Here, we determine high-resolution structures of Myo7a and Myo7b C-terminal MyTH4-FERM domain (MF2) and unveil how they recognize harmonin using a novel binding mode. Systematic definition of interactions between domains of the tripartite complex elucidates how the complex assembles and prevents possible self-association of harmonin-a. Several Myo7a deafness mutants that map to the surface of MF2 disrupt harmonin binding, revealing the molecular basis for how they impact the formation of the tripartite complex and disrupt mechanotransduction. Our results also suggest how switching between different harmonin isoforms can regulate the formation of networks with Myo7a motors and coordinate force sensing in stereocilia.

[1] Structural Motility, Institut Curie, PSL Research University, CNRS, UMR 144, F-75005 Paris, France. [2] Sorbonne Universités, UPMC Univ Paris 06, Sorbonne Universités, IFD, 4 Place Jussieu 75252 Paris, France. [3] Ecole Normale Supérieure, PSL Research University, CNRS, INSERM, Institut de Biologie de l'École Normale Supérieure (IBENS), 75005 Paris, France. [4] Department of Genetics, Cell Biology, and Development, University of Minnesota, Minneapolis, Minnesota 55455, USA. * These authors contributed equally to this work. ** These authors jointly supervised this work. Correspondence and requests for materials should be addressed to A.H. (email: anne.houdusse@curie.fr).

Mechanotransduction is a process by which cells convert mechanical stimuli into electrochemical signals. Special classes of cellular protrusions, such as the stereocilia of sensory hair cells and microvilli of intestinal epithelial cells, are linked together by cadherins, enabling a coordinated response to extracellular stimuli. Links between the tips and sides of adjacent stereocilia are important for transmitting force to the mechano-electrical transduction (MET) channels that convert sound waves into an electrical signal[1,2]. Similarly, microvilli increase the surface area of epithelial cells for absorption, and are tightly connected at their tips to provide an integrated barrier against resident gut bacteria[3]. The cadherin-based connections between stereocilia or microvilli are essential for the formation of mechanically integrated bundles of these projections that must withstand shear stresses imposed on them. This is achieved by anchoring the cytoplasmic domains of cadherins to the actin cytoskeleton via similar tripartite complexes consisting of a class-VII myosin motor (Myo7) and two modular adaptor proteins[3–5] (Fig. 1).

Stereocilia of increasing height are organized into bundles that are linked together by a variety of connections that undergo dynamic changes during development[6,7]. These links serve to both maintain bundle morphology and enable force transmission. When stimulated, hair cell bundles are deflected and the MET channels open, resulting in $Ca^{2+}$ influx. They then undergo adaptation, a process that involves myosin motors, to reduce the electrical response of the hair bundle, thus preventing saturation and ensuring that the bundle remains sensitive to further stimuli[8,9]. Tip links connect the top of a lower stereocilia, where the MET channel is localized, to the side of an adjacent, upper one; and are thought to contribute to resting tension and regulate adaptation. They appear early in development, initially composed of protocadherin 15 (*PCDH15*) dimers, then mature into a heterodimer of PCDH15 and Cadherin 23 (*CDH23*) homodimers localized to the lower- and upper-tip link density (LTLD and UTLD) in adjacent stereocilia, respectively[8,10].

The cytoplasmic tail of *CDH23* is bound to the PDZ domain scaffolding protein harmonin (*USH1C*), which forms a tripartite

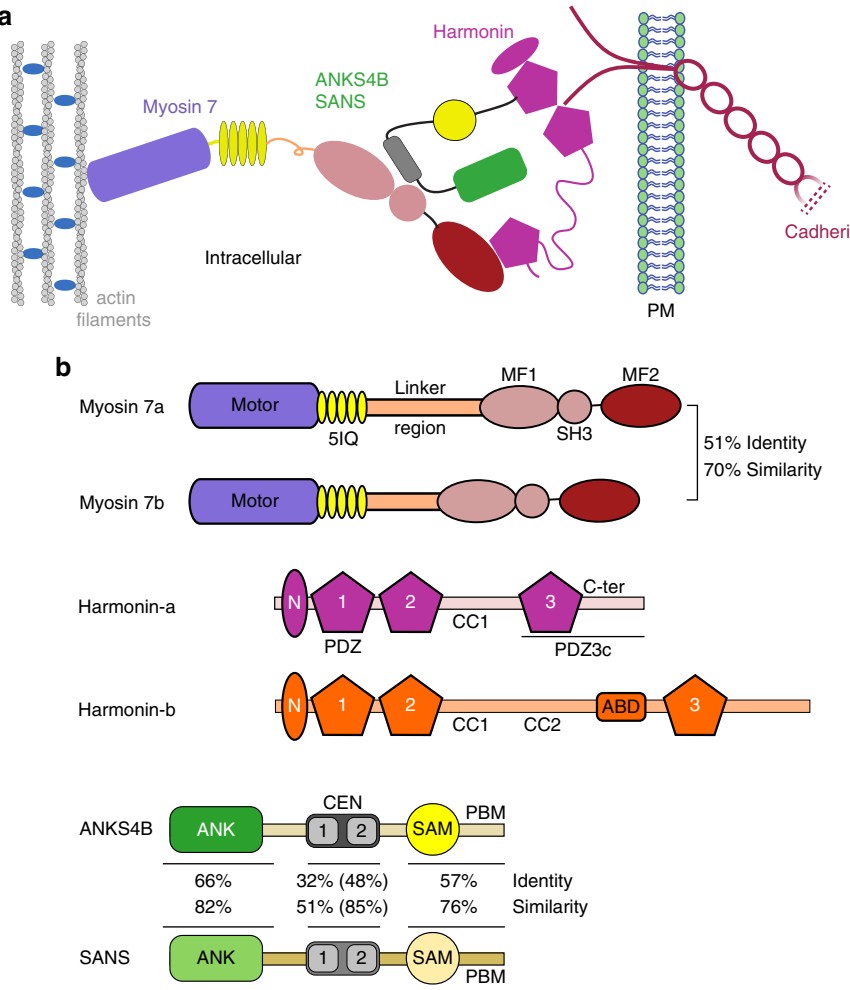

**Figure 1 | Shared components of epithelial cell apical tip link complexes. (a)** Diagram of the ternary, Myo7-based complex that links cadherins to the actin cytoskeleton in stereocilia tip links or microvilli showing the known domain interactions among partners. (**b**) Schematic illustration of Myosin 7a, Myosin 7b, Harmonin-a and -b, ANKS4B and SANS (*USH1G*). Harmonin-a stands for both a1 and a2 isoforms. Harmonin-b represents only the b2 and b3 isoforms. Harmonin-c is not shown. Harmonin-b has an additional predicted coiled-coil region (CC2), an actin binding domain (ABD), and a divergent 47aa long C-terminal extension. Although harmonin isoforms share identical N-ter, PDZ1, 2 and 3, and the CC1, in order to highlight the different functional impact between harmonin-a and -b we represent them in different colours in all figures. The percentage of identity and similarity between the Myo7 tails and between the individual ANKS4B and SANS domains are shown. The CEN domain of ANKS4B/SANS is comprised of two highly conserved CEN regions (labelled 1 and 2) connected by a short linker (see also Fig. 3a). The per cent identity and similarity when only CEN1 and CEN2 regions are taken into account are shown in parentheses.

complex with the ankyrin-repeat SANS adaptor (*USH1G*) that, in turn, binds to the tail domain of Myo7a (refs 4,11) (Fig. 1a). Altogether these proteins constitute a dynamic Usher protein complex that is essential in the morphogenesis of the stereocilia bundle in hair cells and in the calycal processes of photoreceptor cells[12]. While Myo7a and harmonin are found to localize along the length of stereocilia, all three Usher proteins are also concentrated at UTLD, where they associate with the tip link and regulate the function of MET[13–15]. Several mutations in these proteins and/or cadherin are found to be associated with non-syndromic deafness DFNB and DFNA[7], while others result in Usher syndrome type I (USH1), the most severe form of deaf-blindness characterized by profound congenital hearing loss and a prepubertal onset of *retinis pigmentosa*[4]. Similar interactions between homologous proteins connect cadherins to actin filaments in microvilli. Protocadherin 24 (*CDHR2*) links adjacent microvilli via its interaction with the mucin-like protocadherin (*CDHR5*) and both cadherin tails bind to harmonin. ANKS4B, a protein homologous to SANS, also binds to harmonin and the Myo7b tail[3,5] similar to SANS (Fig. 1a). Mutations in, or deletions of, any component of this tripartite complex results in disorganization of apical microvilli[3,5,16].

Myo7 is a crucial component of the tripartite complex that maintains the necessary mechanical tension across the cadherin links[16–18]. It is also implicated in the transport of the complex to the tips of stereocilia or microvilli[4,16,19,20]. The Myo7 tail domain consists of two MyTH4-FERM (MF; myosin tail homology 4; band 4.1, ezrin, radixin, moesin) domains separated by an SH3 domain (Fig. 1b). The N-terminal MF domain (MF1) binds to the CEN domain of SANS/ANKS4B, which can be further divided into CEN1 and CEN2 regions (Fig. 1b). Structures of the MF1 in complex with the CEN domain have been determined for both Myo7a-SANS and Myo7b-ANKS4B (refs 21,22). While the binding sites show a high degree of conservation, the affinities of the two interactions drastically differ, with Myo7a-SANS reported to be notably stronger[22].

There are three major harmonin splice isoforms (a, b and c), each with identical N-terminal domain (NTD), two PDZ domains followed by a coiled-coil region (CC). Harmonin-a and -b have a third PDZ domain; and harmonin-b has an extra CC (CC2), an actin-binding domain (ABD), and an extended C-terminal tail (Fig. 1b). The common NTD and PDZ1 (NPDZ1) domains interact with the SAM-PBM (PDZ binding motif) domains of SANS/ANKS4B (refs 5,21,22) (Fig. 1a,b), while the PDZ2 and/or NTD bind cadherin tails[23–25]. In addition, the extended PDZ3c (PDZ3 plus its C-terminal peptide) of harmonin-a binds directly to the C-terminal MF domain (MF2) of Myo7b but only very weakly to Myo7a, suggesting a possible mechanism for reinforcing the ANKS4B-Myo7b interaction in microvilli[22]. The nature and significance of this interaction and the basis for the specificity between Myo7 paralogs are unknown.

The linker complexes in stereocilia and microvilli have many similarities yet notable differences have been reported, suggesting that they may differ in their assembly and/or maintenance[3,5,16,21,22]. Understanding how cadherin-based linkages between stereocilia and microvilli are generated and maintained requires a complete description of the interactions between Myo7, harmonin and SANS/ANKS4B proteins, and how their modular complexes may switch the motor function from a transporter to a force sensor/anchor. Harmonin plays a pivotal role due to its ability to interact with cadherin as well as all other components of the complex. Gaining structural insights into these cadherin-based complexes is also critical to learn about how they respond to and transduce external forces across cells.

We describe here a combined structural and biochemical approach to define the interactions between Myo7 MF domains and their partners, harmonin and SANS/ANKS4B.

High-resolution structures reveal a conserved mechanism by which Myo7 MF2 domain recognizes harmonin, as well as the molecular basis for how several deafness mutations may disrupt stereocilia mechanotransduction. The results highlight how switching of harmonin isoforms could trigger the formation of distinct complexes that may control the functional role of the motor in cells.

## Results

**Harmonin PDZ3c recognition by Myo7b MF2**. The basis of the harmonin-a PDZ3c/Myo7b MF2 interaction was determined by solving the crystal structures of Myo7b MF2 and of PDZ3c/Myo7b MF2 complex at 2.44 and 1.88 Å resolution, respectively (Tables 1 and 2; Fig. 2a; Supplementary Fig. 1a). The MF2 domain adopts a canonical MyTH4-FERM supramodule, with limited flexibility between the MyTH4 and FERM domains; while the FERM domain consists of three globular subdomains or 'lobes', similar to previously described MF structures[21,22,26–28] (Supplementary Table 1). The majority of the MF2 and PDZ3c interactions occur between the FERM domain and harmonin-a's C-terminal polypeptide chain ($_{545}$YDDELTFF$_{552}$, hereafter referred to as Cter) (Fig. 2b,d), with relatively few contacts between the F1 and F3 lobes of MF2 and the PDZ3 domain (Supplementary Fig. 1d). Limited structural changes are found in the Myo7b MF2 supramodule upon PDZ3c binding (Fig. 2c), with an overall r.m.s.d. of 0.63 Å for 414 Cα atoms between the bound and free structures. This limited conformational pliability upon cargo binding indicates that the groove of a particular FERM domain defines its cargo recognition.

The interacting groove within the FERM domain and the orientation of the Cter peptide in our PDZ3c/Myo7b MF2 structure are notably different from the CEN1 peptide binding to the Myo7 MF1 domain[21,22] (Supplementary Fig. 2d). The distinct orientation of the FERM lobes in Myo7b MF1 and MF2 domains

**Table 1 | Details of proteins and their boundaries used in the study.**

| Construct name | Residues |
|---|---|
| *Myosin 7a* (Uniprot: Q13402-1)* | |
| M7a MF1.SH3 | 991–1692 |
| M7a MF2 | 1702–2215 |
| | |
| *Myosin 7b* (Uniprot: Q16PIF6-1) | |
| M7b MF1.SH3 | 958–1586 |
| M7b MF2 | 1605–2116 |
| | |
| *Harmonin-a1* (*USH1C*) (Uniprot: Q9Y6N9-1) | |
| NPDZ1 | 1–197 |
| NPDZ1.PDZ2 | 1–301 |
| PDZ2-end | 194–552 |
| PDZ3c | 428–552 |
| PDZ3c$_{\Delta Cter}$ | 428–542 |
| Full length (FL)-harmonin | 1–552 |
| FL-harmonin-ΔCter | 1–542 |
| | |
| *Harmonin-b3* (Uniprot: Q9Y6N9-5) | |
| PDZ3c$_{isoB}$ | 728–899 |
| | |
| *SANS* (*USH1G*) (Uniprot: Q495M9) | |
| SANS$_{CEN-PBM}$ | 300–461 |
| | |
| *ANKS4B* (Uniprot: Q8N8V4) | |
| ANKS4B$_{CEN-PBM}$ | 259–417 |

*Note that residues 1523-1560 were deleted from MF1.

**Table 2 | Data collection and refinement statistics.**

|  | Myo7b MF2 Native | Myo7b MF2 SeMet | Myo7b MF2 + HarA PDZ3c | Myo7a MF2 + HarA PDZ3c |
|---|---|---|---|---|
| *Data collection* |  |  |  |  |
| Space group | C 2 | C 2 | P 2₁ | C 2 2 2₁ |
| Cell dimensions |  |  |  |  |
| $a, b, c$ (Å) | 123.7, 42.8, 118.4 | 124.3, 42.7, 118.3 | 69.62, 42.56, 118.17 | 121.52, 151.71, 100.8 |
| $\alpha, \beta, \gamma$ (deg) | 90.0, 97.7, 90.0 | 90.0, 97.8, 90.0 | 90.0, 98.02, 90.0 | 90.0, 90.0, 90.0 |
| Resolution (Å) | 50–2.44 (2.53–2.44)* | 50–2.86 (3.11–2.86) | 50–1.88 (1.95–1.88) | 50–2.60 (2.70–2.60) |
| $R_{sym}$ (%) | 7.2 (61.8) | 17.9 (68.7) | 9.7 (124.3) | 5.3 (53.2) |
| $I/\sigma$ | 11.86 (1.87) | 8.42 (1.65) | 12.77 (1.0) | 14.16 (1.68) |
| $CC_{1/2}$ (%) | 99.7 (91.5) | 99.1 (84.9) | 99.8 (54.3) | 99.8 (76.2) |
| Completeness (%) | 97.4 (93.7) | 99.5 (95.0) | 95.0 (61.0) | 97.0 (96.0) |
| Redundancy | 3.4 (3.1) | 14.0 (8.1) | 6.4 (4.0) | 3.1 (3.0) |
| Wavelength (Å) | 0.9786 | 0.9792 | 0.9060 | 0.9786 |
|  |  |  |  |  |
| *Refinement* |  |  |  |  |
| Resolution (Å) | 35.17–2.44 |  | 34.47–1.88 | 19.84–2.60 |
| No. reflections | 22,553 (2,098) |  | 53,484 (3,181) | 87,935 (8,118) |
| $R_{work}$ / $R_{free}$ (%) | 20.86/25.80 |  | 18.98/21.90 | 17.97/23.70 |
| No. atoms |  |  |  |  |
| Protein | 4,026 |  | 4,789 | 4,737 |
| Ligand/ion | 37 |  | 37 | 8 |
| Water | 139 |  | 380 | 328 |
| R.m.s.d.'s |  |  |  |  |
| Bond lengths (Å) | 0.010 |  | 0.010 | 0.010 |
| Bond angles (°) | 1.10 |  | 0.99 | 1.13 |
| Ramachandran plot |  |  |  |  |
| Favoured (%) | 97.73 |  | 97.62 | 96.27 |
| Accepted (%) | 100 |  | 99.66 | 100 |
| Outliers (%) | 0 |  | 0.34 | 0 |
| PDB Code | 5MV7 |  | 5MV8 | 5MV9 |

*Values in parentheses are for highest-resolution shell.

(Supplementary Movie 1) readily explains their specific recognition of different targets within their FERM core elements (reviewed in[27]). The five linkers in the FERM lobes (Supplementary Figs 2e,f and 3) dictate the orientation of each lobe and thus define the shape and, as a result, the specificity of the central groove. While the binding mode of harmonin PDZ3c to Myo7b FERM has some similarity to that of Heg1 tail bound to the KRIT FERM domain[29] (Supplementary Fig. 2d, right), the positioning and specific recognition of the two tails drastically differ. It is worth noting that Myo15a's MF-containing tail also binds to the PDZ domain protein whirlin, however this interaction is mediated by a class-I PBM in Myo15a binding to the PDZ3 domain of whirlin. Thus, Myo7b MF2 recognizes harmonin PDZ3c via a new binding mode.

Binding of PDZ3c to MF2 results in a buried solvent-accessible surface area of 1276 Å² (484 Å² from PDZ3 domain and 792 Å² from Cter). The Cter fits into the groove between the three FERM lobes (Fig. 2b, Supplementary Fig. 1c) and interacts with all three via numerous, mainly hydrophobic, interactions. Two aspartates, D546 and D547, are involved in hydrogen bonds with Myo7b R1921 and S2082 (Fig. 2d). The last two phenylalanines of the Cter reach to the centre of the cloverleaf where they interact with all three lobes through extensive hydrophobic and hydrogen bond contacts (Fig. 2d). In this buried environment, the negatively charged carboxylic group of the Cter interacts specifically with three residues of Myo7b, K1918, Q1914 and W1895 (Fig. 2d). The harmonin-a Cter peptide is highly conserved in vertebrates (Supplementary Fig. 5), with the most important residues recognized by the FERM domain forming the following consensus sequence: φ-D-D-x-Ψ-x-FF-COOH, where φ and Ψ indicate Phe/Tyr and Leu/Val/Ile/Met, respectively (Fig. 2b,d). Notably, the canonical class-II PBM binding site in the PDZ3

domain[24,30,31] is not involved in binding to Myo7b MF2 and is available for other partners (Fig. 2c, Supplementary Fig. 2c).

The extensive interactions found between the PDZ3c and the FERM domain (Fig. 2d) are consistent with Myo7b MF2 and harmonin PDZ3c forming a tight complex ($K_d$ of 1.4 μM; Table 3, Supplementary Fig. 4a), as previously reported[22]. The harmonin PDZ3c mutants, F551V-F552V and D546R, exhibit substantially reduced affinity for MF2, $K_d$ of 68 and > 300 μM, respectively (Table 3, Fig. 2b, Supplementary Fig. 4d). A triple mutant D546R-F551V-F552V, as well as deletion of the entire Cter motif (ΔCter) abolishes the interaction (Table 3, Supplementary Fig. 4b). Finally, mutations of the Myo7b MF2 (L2083W and R1921E-F1923V) that disrupt major contacts in the interface also abolish binding (Table 3, Fig. 2b, Supplementary Fig. 4c). Altogether, our results establish that the conserved harmonin-a Cter motif mediates binding to Myo7b MF2 and highlight the importance of extensions of the PDZ domains to specifically mediate interactions with binding partners (Supplementary Fig. 2a,b)[32–34].

**Conserved mechanism of recognition by Myo7s.** The MF2 of Myo7a was reported to interact weakly, if at all, with PDZ3c[22]; and this could be attributed to divergent sequences (Fig. 1b) and/ or differences in the orientation of the Myo7a MF2 lobes. All of the residues present in the central FERM groove of Myo7b MF2 involved in binding to PDZ3c are conserved in both vertebrate paralogs, except for a small Tyr2026[Myo7a]/Phe1923[Myo7b] difference (Supplementary Fig. 6, arrows and #). Consistent with this sequence conservation, PDZ3c binds to Myo7a MF2 with ~ 1 μM affinity, the same affinity as the Myo7b MF2/PDZ3c interaction (Table 3, Supplementary Fig. 4d). This is in contrast to initial reports that did not detect a significant interaction[22],

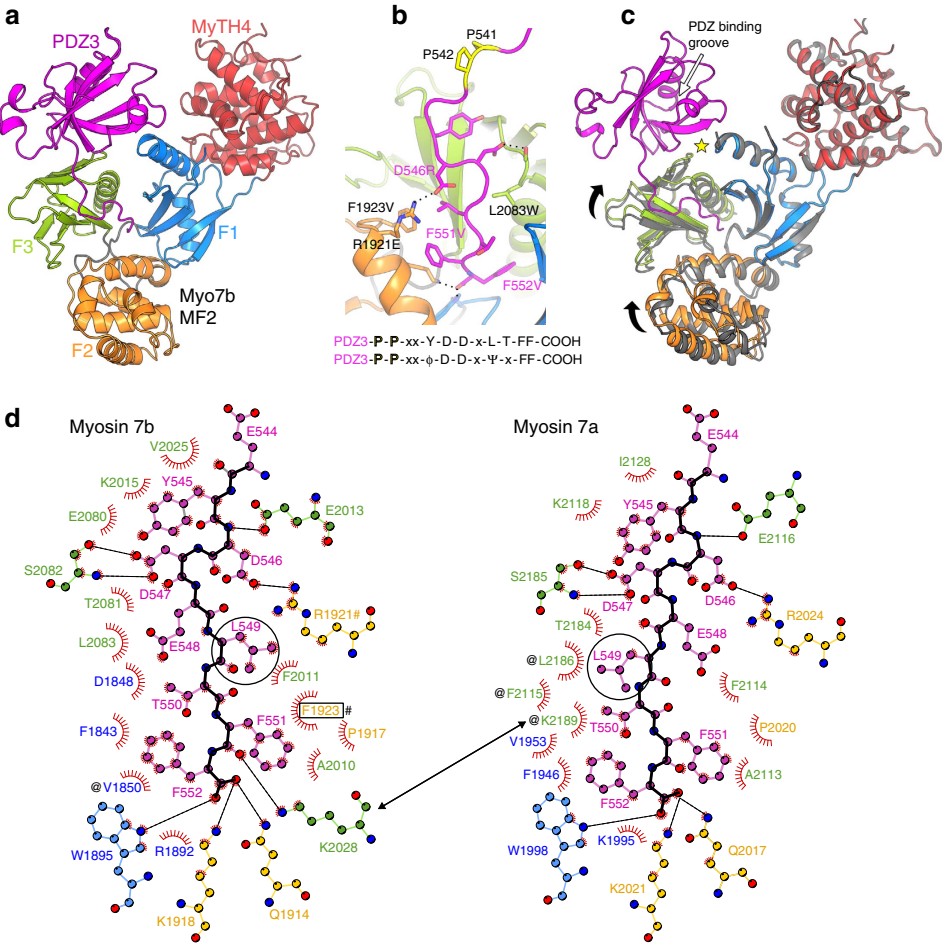

**Figure 2 | Structural basis of harmonin PDZ3c recognition by Myo7b MF2.** (**a**) Structure of the harmonin PDZ3c-Myo7b MF2 complex. The MyTH4 domain is shown in red and the FERM subdomains are coloured in blue (F1), orange (F2) and green (F3) throughout the figures. The harmonin PDZ3c is shown in magenta. (**b**) Critical residues mediating interactions between the harmonin Cter extension (magenta) and the Myo7b FERM (ribbon representation) domain. Important side chains are shown in sticks. Two conserved prolines (P541 and P542) at the end of the PDZ3 domain help direct the Cter extension. The motif that mediates binding to the FERM domain is shown (bottom). (**c**) Superimposition of the free (dark grey) and PDZ3c-bound (multi-coloured) Myo7b MF2 structures using the F1 lobe as a reference. The black arrows indicate the lobe displacements upon PDZ3c binding. The white arrow in the PDZ3 domain indicates the partner-binding groove within the PDZ3 domain. The yellow star highlights the interface between the PDZ3 domain and the Myo7b F1 and F3 lobes. The minor changes in the orientation of the three lobes (<5°) promote optimal interactions between the Cter peptide and the FERM domain via an induced fit of small amplitude. (**d**) Diagram of the interactions between the harmonin PDZ3 Cter extension (magenta, black backbone) and the Myo7 FERM domain. Dotted lines: hydrogen bond within distances <3.5 Å; Red spikes: hydrophobic interactions. The major difference between the two complexes is in the orientation of harmonin L549 side chain (black circles) due to the F1923$^{Myo7b}$ (black box)/Y2026$^{Myo7a}$ substitution. The following residue, T550, also changes its environment (# denotes residues interacting with L549, while @ indicates residues interacting with T550). The black arrow indicates the homologous lysines 2086/2189 (Myo7b/Myo7a) that interact differently with PDZ3c. This figure is generated using LigPlot+$^{69}$.

despite the fact that the binding assays were performed at the same ionic strength (100 mM NaCl). The Cter motif is also critical for the Myo7a MF2/PDZ3c interaction, since the PDZ3$_{D546R-F551V-F552V}$ and ΔCter mutants abolish binding (Table 3). These findings suggest that the Myo7a and Myo7b MF2 domains use a similar binding mode to interact with PDZ3c.

The structure of Myo7a MF2 bound to harmonin PDZ3c was determined at 2.6 Å resolution (Table 2; Supplementary Fig. 1e). Comparison with the PDZ3c/Myo7b MF2 structure results in an r.m.s.d. of 1.20 Å (for 411 Cα atoms) (Supplementary Fig. 1g) despite the low-sequence identity (51%) between the two MF2 domains and major differences in the crystal packing environments. The harmonin PDZ3c binds in the same overall manner by inserting the Cter motif into Myo7a MF2's central FERM cavity (Supplementary Fig. 1g). However, the Cter motif adopts a slightly different conformation in the two complexes

(Supplementary Fig. 1f), namely residues Leu549 and Thr550 interact distinctly in the two FERM cavities (Fig. 2d) due to the Tyr2026$^{Myo7a}$/Phe1923$^{Myo7b}$ variation (Fig. 2d; supplementary Fig. 1f). Such slight adaptability of the FERM groove results mainly from conformational variability in the F1 lobe (Supplementary Fig. 1g,h). The affinities of PDZ3c for either Myo7a or Myo7b MF2 are similar despite these changes (Table 3), showing that some adaptation of the binding mode in this FERM cavity can occur without much cost in affinity.

There are relatively few contacts between the surface of the PDZ3 domain and the F1 and F3 lobes of Myo7 MF2 (Supplementary Fig. 1d). The PDZ3 E469A mutant, which disrupts the interactions with two main chain nitrogens of the F3 lobe (Supplementary Fig. 1d, yellow star), modestly reduces the binding affinity for both Myo7s' MF2s (from $K_d$ of 1.6–1.8 μM to ~5–6 μM; Table 3). Introducing a bulky residue at the interface,

**Table 3 | Structural basis for harmonin PDZ3c recognition by Myo7b MF2.**

| Myosin | Harmonin | $K_d$ |
|---|---|---|
| *Myo7b MF2 binding to Harmonin* | | |
| M7b MF2 | PDZ3c (isoA) | $1.4 \pm 0.6\,\mu M$ ($n = 2$)*,† |
| | $^{YFP}$PDZ3c (isoA) | $1.9 \pm 0.6\,\mu M$ ($n = 2$) |
| M7b MF2 L2083W | $^{YFP}$PDZ3c (isoA) | w.b. ($> 300\,\mu M$) ($n = 2$) |
| M7b MF2 R1921E-F1923V | PDZ3c (isoA) | n.b.* |
| | $^{YFP}$PDZ3c (isoA) | w.b. ($> 100\,\mu M$) ($n = 2$) |
| $^{YFP}$M7b MF2 | PDZ3c (isoA) F551V-F552V | w.b. ($> 100\,\mu M$) ($n = 2$) |
| $^{YFP}$M7b MF2 | PDZ3c (isoA) D546R | w.b. ($> 300\,\mu M$) ($n = 2$) |
| $^{YFP}$M7b MF2 | PDZ3c (isoA) D546R-F551V-F552V | n.b. |
| M7b MF2 | PDZ3c (isoA) D546R-F551V-F552V | n.b.* |
| $^{YFP}$M7b MF2 | PDZ3 ΔCter | n.b. |
| M7b MF2 | $^{YFP}$FL (isoA) | $2.05 \pm 0.79\,\mu M$ ($n = 2$) |
| $^{YFP}$M7b MF2 | PDZ2-end (isoA) | $2.79 \pm 0.79\,\mu M$ ($n = 2$) |
| $^{YFP}$M7b MF2 | NPDZ1.PDZ2 | n.b. |
| $^{YFP}$M7b MF2 | FL ΔCter | n.b. |
| M7b MF2 | PDZ3c (isoB) | n.b.* |
| | | |
| *Myo7a MF2 binding to Harmonin* | | |
| M7a MF2 | PDZ3c (isoA) | $1\,\mu M$* |
| $^{YFP}$M7a MF2 | PDZ3c (isoA) | $1.4 \pm 0.2\,\mu M$ ($n = 2$) |
| $^{YFP}$M7a MF2 | PDZ3c (isoA) F551V-F552V | n.b. |
| $^{YFP}$M7a MF2 | PDZ3c (isoA) D546R | n.b. |
| $^{YFP}$M7a MF2 | PDZ3c (isoA) D546R-F551V-F552V | n.b. |
| $^{YFP}$M7a MF2 | PDZ3c ΔCter | n.b. |
| M7a MF2 | $^{YFP}$FL (isoA) | $2.36\,\mu M$ |
| $^{YFP}$M7a MF2 | PDZ2-end (isoA) | $1.88\,\mu M$ |
| $^{YFP}$M7a MF2 | NPDZ1.PDZ2 | n.b. ($n = 2$) |
| $^{YFP}$M7a MF2 | FL ΔCter | n.b. |
| M7a MF2 | PDZ3c (isoB) | n.b.* |
| | | |
| *Harmonin PDZ3 contribution to the binding of Myo7 MF2* | | |
| M7a MF2 | $^{YFP}$PDZ3c (isoA) E469A | $5.4 \pm 1.5\,\mu M$ ($n = 2$) |
| M7a MF2 | $^{YFP}$PDZ3c (isoA) I476W | $17.8 \pm 1.1\,\mu M$ ($n = 2$) |
| M7a MF2 | $^{YFP}$PDZ3c (isoA) E469A-I476W | $70.4 \pm 7.2\,\mu M$ ($n = 2$) |
| M7b MF2 | $^{YFP}$PDZ3c (isoA) E469A | $6.11 \pm 0.44\,\mu M$ ($n = 2$) |
| M7b MF2 | $^{YFP}$PDZ3c (isoA) I476W | $18.1 \pm 1.4\,\mu M$ ($n = 2$) |
| M7b MF2 | $^{YFP}$PDZ3c (isoA) E469A-I476W | $38.1 \pm 4.9\,\mu M$ ($n = 2$) |

n.b. and w.b. stand for no binding (no fit possible) and weak binding (the end of the curve is missing, and only a rough estimate of the $K_d$ can be provided), respectively.
*ITC results; $n$ = number of independent experiments.
†Previously measured 1.7 μM (ref. 22).

I476W, results in even weaker binding ($K_d$ of ∼18 μM; Table 3), likely due to its interference with key interactions between the PDZ3 and the FERM domain (Supplementary Fig. 1d, yellow star). Thus, PDZ3 contributes modestly to the interaction with MF2. The Myo7a MF2 domain has been shown by pull-down experiment to bind to the NPDZ1.PDZ2 domains of harmonin[23], suggesting that Myo7a could have multiple direct interactions with this adaptor. However, MST binding assays failed to detect any direct interaction between NPDZ1.PDZ2 and the MF2 of either Myo7a or Myo7b (Table 3). Although the different techniques and/or conditions used for those binding assays might lead to the conflicting outcomes, consistent with our results, the ΔCter full-length (FL) harmonin-a does not interact with either Myo7a or Myo7b MF2 (Table 3). In addition, the FL-harmonin and the PDZ2-end region (Table 1) both bind to either MF2 with a similar affinity as PDZ3c (Table 3), suggesting that PDZ3c mediates the main interaction between the MF2 and harmonin-a. Taken together, these results demonstrate that the binding mode of MF2 to PDZ3c is conserved between Myo7 paralogs and involves both an essential, tight association with the harmonin Cter motif, as well as critical interactions with the core PDZ3 domain. Moreover, the tight binding between the Cter of harmonin-a and Myo7a MF2 suggests a potential, previously overlooked role for harmonin-a in stereocilia.

**Interactions within the Myo7 tripartite complex.** A number of binary interactions between members of the Myo7a and Myo7b tripartite complexes have been characterized. While many of these are conserved between the two homologous systems, there are significant differences[3,5,21,22]. An intriguing divergence is found between the interactions of Myo7a-SANS and Myo7b-ANKS4B. Neither ANKS4B CEN nor SANS CEN domain was reported to bind to Myo7a or Myo7b MF1.SH3, respectively[22], suggesting the homologous interactions are not interchangeable. However, structures of the two complexes have shown that they share an almost identical interaction site[21,22] and the sequences of the ANKS4B and SANS CEN1 and CEN2 regions are highly conserved (85%, Figs 1 and 3a). In agreement with the structure and sequence conservation involved in the interactions, $SANS_{CEN-PBM}$ and $ANKS4B_{CEN-PBM}$ (including both CEN regions and the SAM domain; Fig. 1 and Table 1) both bind to Myo7b MF1.SH3 with similar affinities ($K_d$ of 3–6 μM) (Fig. 3c, Table 4). Moreover, $ANKS4B_{CEN-PBM}$ also binds Myo7a MF1.SH3 with a comparable affinity ($K_d$ of ∼5 μM) (Fig. 3c, Table 4). Mutation of conserved surface residues (A1128E-R1129E-K1192E; Fig. 3b) in the $Myo7b_{MyTH4}$ CEN2-binding site abrogates both $ANKS4B_{CEN-PBM}$ and $SANS_{CEN-PBM}$ binding (Table 4), consistent with a conserved CEN2-binding site on Myo7b MF1 for both ANKS4B and

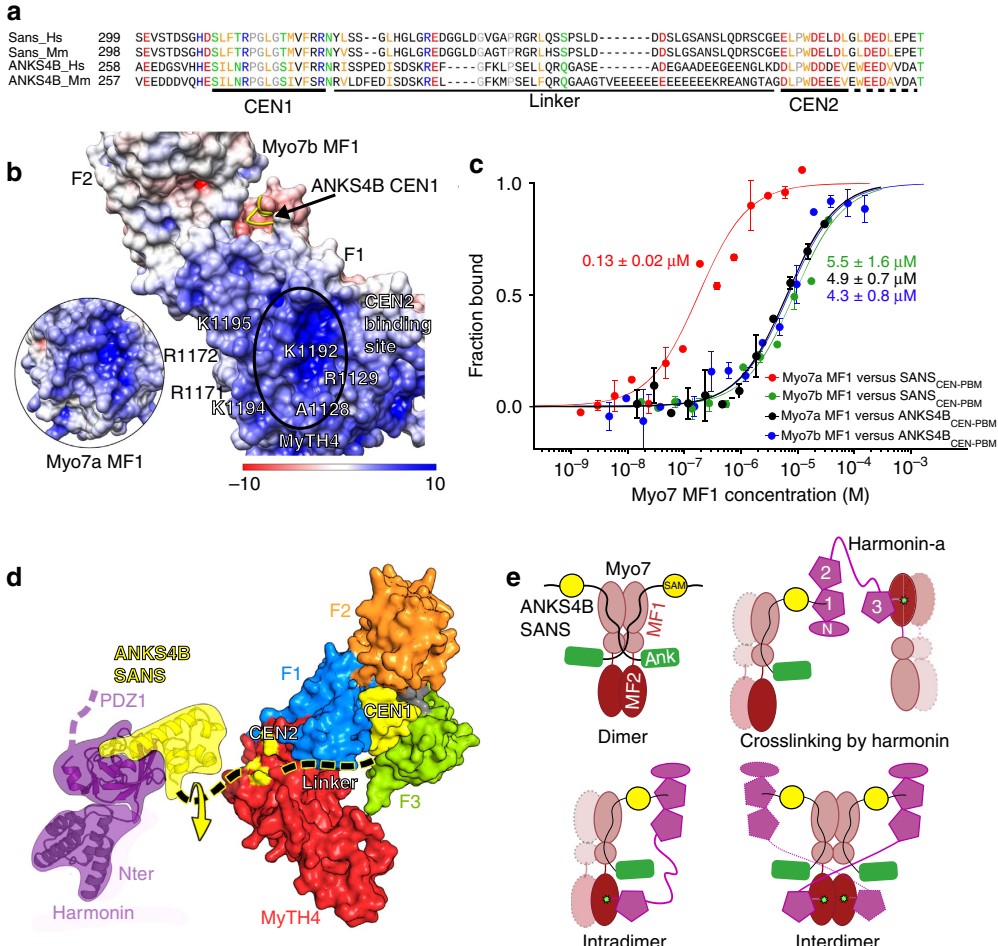

**Figure 3 | Myosin 7 MF1 binding to SANS$_{CEN-PBM}$ and ANKS4B$_{CEN-PBM}$.** (**a**) Alignment of SANS and ANKS4B CEN domain (Hs: human; Mm: mouse). Conserved residues are shown in red (negatively charged), blue (positively charged), yellow (hydrophobic), green (polar and not charged) and grey (proline and glycine). (**b**) Electrostatic surface potential of Myo7b MF1/ANKS4B (5F3Y) and Myo7a MF1/SANS (3PVL) (inset). The residues found in the positively charged pocket in the MyTH4 domains that participate in SANS or ANKS4B CEN2 binding are indicated (note that the CEN2 region is highly negatively charged, (**a**)). The surface potential was calculated using the APBS server with default parameters. (**c**) MST curves comparing the Myo7 MF1.SH3 binding to either SANS$_{CEN-PBM}$ or ANKS4B$_{CEN-PBM}$. Averaged $K_d$s from two technical replicates (mean ± s.d.) are shown. (**d**) Model of the tripartite Myo7 MF1-SANS/ANKS4B-Harmonin complex. The Myo7 MF1 is shown in a surface representation, with the binding site of CEN2 highlighted in yellow. CEN1 and SAM of ANKS4B or SANS are also shown in yellow. The linkers are shown as black dashed line. The short variable linker between CEN2 and SAM domain is likely flexible, allowing for movement of the SAM domain relative to the Myo7 tail (yellow arrow). (**e**). Cartoon models showing the different tripartite complexes that could be assembled. Full length SANS can dimerize via a region between the ANKyrin domain and CEN1 (ref. 35) and, thus, potentially facilitates Myo7 dimerization (dimer). While interacting with MF1 (brown) via SANS/ANKS4B (yellow and green), the long CC1 region of harmonin may allow PDZ3c to reach a MF2 of another Myo7 molecule (crosslinking), the same Myo7 molecule (intradimer), or the other MF2 in the same myosin dimer (interdimer).

SANS. In contrast, SANS$_{CEN-PBM}$ binds Myo7a MF1 with 130 nM $K_d$ (Table 4), an affinity much higher than for Myo7b as previously reported[21], revealing how small changes in sequence can modulate the strength of association between complex members. Interestingly, the presence of saturating concentrations of harmonin NPDZ1.PDZ2 does not affect the binding between MF1 and SANS/ANKS4B (Table 4), establishing that SANS$_{CEN-PBM}$/ANKS4B$_{CEN-PBM}$ binding to harmonin does not influence their interactions with Myo7. Furthermore, size-exclusion chromatography combined with multi-angle light scattering (SEC-MALS) demonstrates that the stoichiometry of the Myo7a MF1/SANS$_{CEN-PBM}$ and MF1/SANS$_{CEN-PBM}$/NPDZ1 complexes are 1:1 and 1:1:1, respectively (Supplementary Fig. 7a,b). Altogether, these results establish that the tripartite complex can form based on the known, characterized binary interactions, where SANS/ANKS4B binds to the MF1 via their

CEN1 and CEN2 regions and to harmonin NPDZ1 via their SAM-PBM domain (Fig. 3d).

The defined interactions between members of the tripartite complex and their intrinsic plasticity suggest multiple, possible ways they can assemble. The linker between CEN2 and SAM is long enough (∼10 aa) to provide significant rotational flexibility for the orientation of MF1 relative to harmonin and SANS/ANKS4B (Fig. 3d). In addition, the region upstream of CEN1 in SANS has been shown to dimerize[35], thus binding to SANS may promote Myo7 dimerization (Fig. 3e, dimer). On the basis of all the current binding and structural data, we propose several possible models for how the tripartite complex may assemble (Fig. 3e). A single-harmonin molecule, through interactions with both Myo7 tail's MF1 (via SANS/ANKS4B) and MF2 domains, can either crosslink two independent Myo7 (Fig. 3e, crosslinking), bind to the same Myo7 (Fig. 3e,

**Table 4 | Myo7 MF1.SH3 interaction with SANS or ANKS4B.**

| Myosin/Harmonin | SANS/ANKS4B | $K_d$ |
|---|---|---|
| M7b MF1.SH3 | $^{YFP}SANS_{CEN-PBM}$ | $5.4 \pm 1.6\,\mu M$ ($n = 2$) |
| M7a MF1.SH3 | $^{YFP}SANS_{CEN-PBM}$ | $0.13 \pm 0.02\,\mu M$ ($n = 3$)* |
| M7a MF1.SH3 | $^{YFP}SANS_{CEN-PBM}$ + NPDZ1.PDZ2 | $0.386\,\mu M$ |
| NPDZ1.PDZ2 | $^{YFP}SANS_{CEN-PBM}$ | $0.017\,\mu M$ |
| M7b MF1.SH3 A1128E | $^{YFP}SANS_{CEN-PBM}$ | $18.2 \pm 2.8\,\mu M$ ($n = 2$) |
| M7b MF1.SH3 A1128E-R1129E-K1192E | $^{YFP}SANS_{CEN-PBM}$ | n.b. ($n = 2$) |
| M7b MF1.SH3 K1194E-K1195E R1171E-R1172E | $^{YFP}SANS_{CEN-PBM}$ | $6.6 \pm 2.3\,\mu M$ ($n = 2$) |
| M7a MF2 | $^{YFP}SANS_{CEN-PBM}$ | n.b. |
| M7b MF2 | $^{YFP}SANS_{CEN-PBM}$ | n.b. |
| $^{YFP}$M7b MF1.SH3 | ANKS4B$_{CEN-PBM}$ | $2.9 \pm 0.03\,\mu M$ ($n = 2$)* |
| M7b MF1.SH3 | $^{YFP}ANKS4B_{CEN-PBM}$ | $4.3 \pm 0.8\,\mu M$ ($n = 2$) |
| M7a MF1.SH3 | $^{YFP}ANKS4B_{CEN-PBM}$ | $4.9 \pm 0.7\,\mu M$ ($n = 2$) |
| M7b MF1.SH3 | $^{YFP}NPDZ1.PDZ2$ + ANKS4B$_{CEN-PBM}$ | $4.0\,\mu M$ |
| NPDZ1.PDZ2 | $^{YFP}ANKS4B_{CEN-PBM}$ | $0.009\,\mu M$ |
| M7b MF1.SH3 A1128E | $^{YFP}ANKS4B_{CEN-PBM}$ | $13.1 \pm 0.7\,\mu M$ ($n = 2$) |
| M7b MF1.SH3 A1128E-R1129E-K1192E | $^{YFP}ANKS4B_{CEN-PBM}$ | n.b. ($n = 2$) |
| M7b MF1.SH3 K1194E-K1195E R1171E-R1172E | $^{YFP}ANKS4B_{CEN-PBM}$ | $3.1 \pm 0.2\,\mu M$ ($n = 2$) |

$n$ = number of independent experiments. n.b. and w.b. stand for no binding (no fit possible) and weak binding (the end of the curve is missing, and only a rough estimate of the $K_d$ can be provided), respectively.
*Previously measured 0.05 and 1.06 μM, respectively[21,22].

intradimer), or crosslink two Myo7 within a particular dimer (Fig. 3e, interdimer). This modular nature of the Myo7 tail–adaptor interaction could thus allow distinct motor clustering, depending on the presence of various partners and their respective concentrations that may vary in the different environments in stereocilia and microvilli. Dynamic variations in the partners' expression would result in diverse tripartite complex assemblies, as proposed in Fig. 3e, and may control whether Myo7 is used to transport an adaptor complex to target sites (that is, cadherin tails) or to anchor the different cadherin-based linkages during microvilli and stereocilia development.

**Tripartite complex in stereocilia and MF2 deafness mutants.** Multiple harmonin isoforms present in stereocilia likely contribute both to anchoring the tip links and to the stabilization of other inter-stereociliary links[36–38]. Harmonin-b, abundant in stereocilia, has a longer Cter tail lacking the conserved motif required for interaction with the MF2 domain. Indeed, no binding was observed between the PDZ3 domain with the longer extension (PDZ3c$_{isoB}$) and either the Myo7a or Myo7b MF2 (Table 3), establishing that interaction between harmonin and the MF2 domain of Myo7 is dependent on the specific harmonin isoform. Thus, in stereocilia, interactions between harmonin-b and myosin in the tripartite complexes are likely restricted to the indirect, but strong MF1–SANS interactions[21,22]; and the proposed tripartite complex models (Fig. 3e) can only form with harmonin-a containing the short Cter motif, but not harmonin-b (Figs 1b and 2b, Supplementary Fig. 5).

Harmonin-b is an important component of the tip link complex[23,36,39], yet its association with Myo7a would not require the MF2 domain. However, several deafness mutations suggest the importance of this domain for the integrity of stereocilia. In light of the new structures, 25 missense mutations of Myo7a MF2 that cause deafness were mapped onto the structure to assess their potential impact on binding to partners (Fig. 4c,

Supplementary Table 2 and references therein, Supplementary Movie 2). The majority of these likely result in destabilization of the MF2 fold. However, mutations affecting three surface residues (K2118N, G2163S, G1982E/R) suggest a role for the surface next to the F1$_{H2-S5}$ loop and the F3 lobe in making specific interactions with partners. Interestingly, these four mutations significantly weaken the interaction with PDZ3c (Fig. 4), suggesting the importance of the Myo7a MF2/harmonin PDZ3c interactions in hearing and a functional role for harmonin-a. Consistently, a study of mutant mouse Myo7a targeting showed that GFP-Myo7a G1982R failed to localize to stereocilia[20]. Furthermore, a specific deficit in harmonin-b (Ush1c$^{dfcr-2J/dfcr-2J}$ mutant mice) was shown not to impair the stereocilia bundle morphology whereas Ush1c$^{dfcr}$ mutant mice, lacking both harmonin-a and -b, have abnormal stereocilia[37,38]. Altogether, these results support an essential role for the MF2 domain via interactions with harmonin-a and/or other partners despite the fact that the tripartite complex would remain together through strong SANS interactions. Disruption of the interactions involving Myo7a MF2 would be predicted to interfere with the formation of strong stereocilia linkages, resulting in deafness.

**Harmonin autoinhibition and self-association.** Harmonin has a pivotal role in the formation and function of the tripartite complex and has been suggested to interact with itself, possibly promoting formation of a larger anchoring complex[24,35]. Various inter-domain interactions have been reported but have not been well-characterized. The Cter motif involved in binding with the Myo7 MF2 domain has been suggested to contain a putative class-I PBM[24] (Supplementary Fig. 5) that can bind to PDZ1 and PDZ2. Consistently, NPDZ1 binds PDZ3c with a weak affinity ($K_d \sim 10\,\mu M$) (Fig. 5a, Table 5). The affinity between NPDZ1 and PDZ2-end or between NPDZ1.PDZ2 and PDZ3c is slightly higher ($K_d \sim 3$–$5\,\mu M$). The Cter motif is essential for this interaction as PDZ3c$_{\Delta Cter}$ or PDZ3c$_{F551V-F552V}$ abolishes binding (Table 5). Furthermore, no interaction was detected between the

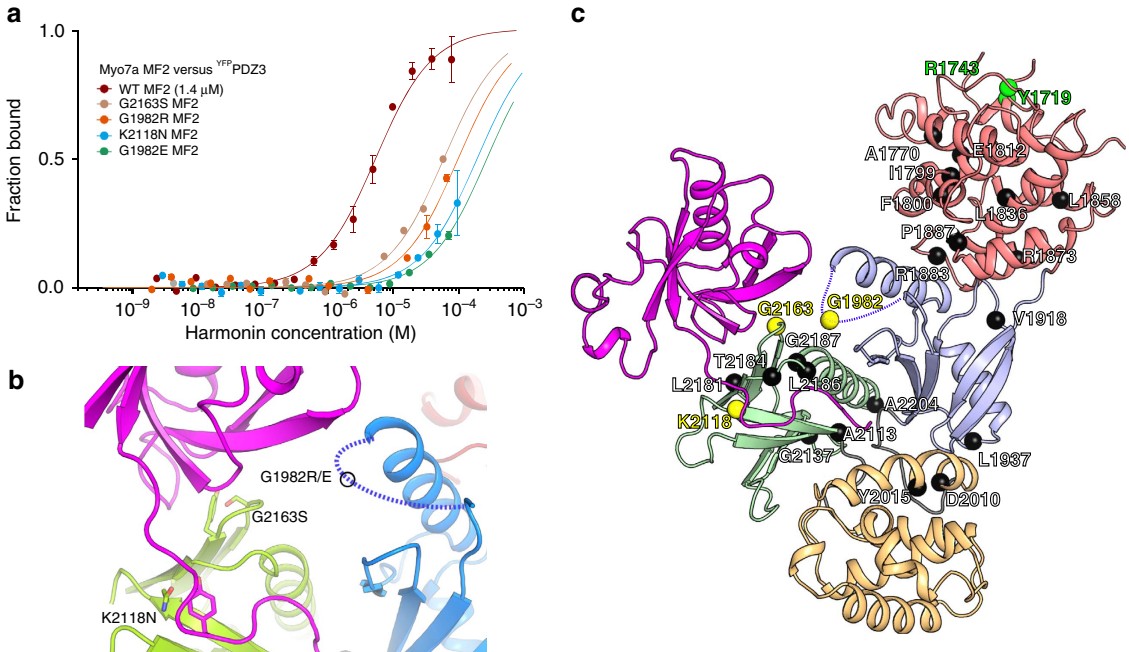

**Figure 4 | Myo7a deafness mutations impair PDZ3c-MF2 interaction.** (**a**) Analysis of the interactions between Myo7a MF2 deafness mutants and PDZ3c by MST. Averaged $K_d$s from two technical replicates (mean ± s.d.) are shown. All four mutants display weak binding affinities (>60 μM) (**b**) The location of the three surface mutations on the F1 (blue) and F3 (green) lobes of MF2 relative to PDZ3c (magenta) is shown. The missing loop in the F1 lobe is indicated by a dashed line. (**c**) The location of known deafness mutations found in Myo7a MF2 is shown in black spheres, with the mutants tested shown in yellow spheres (see also Supplementary Table 2 and Supplementary Movie 2).

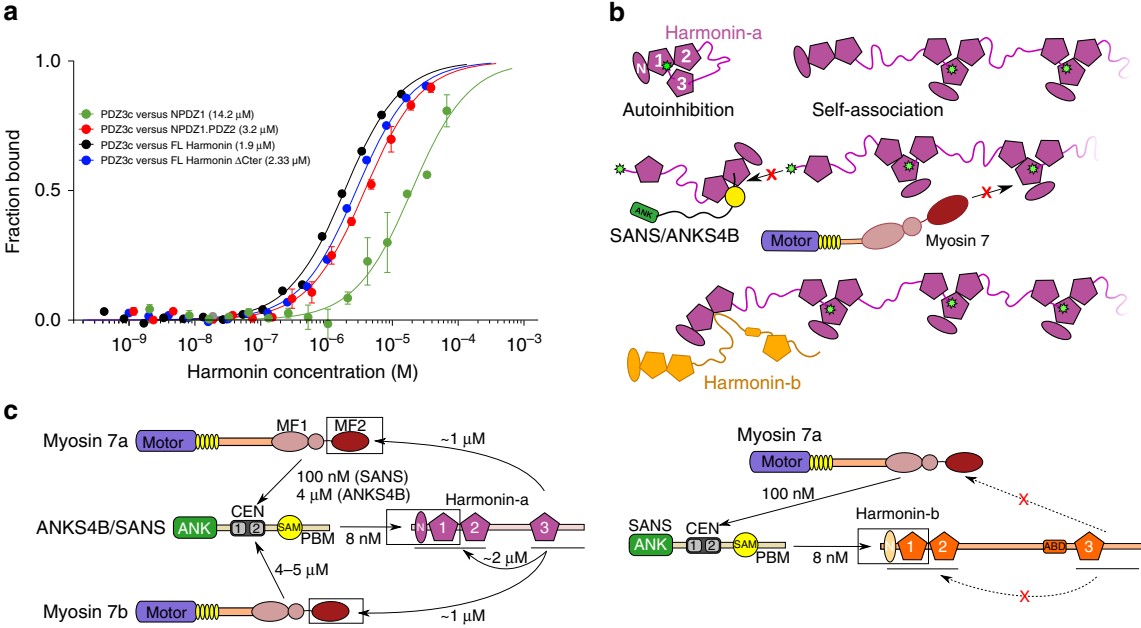

**Figure 5 | Intramolecular interaction between the N- and C-terminus of harmonin and compatibility with Myo7 clustering.** (**a**) Representative MST binding curves measuring the interactions between PDZ3c and NPDZ1, NPDZ1.PDZ2, FL-harmonin ΔCter or FL-harmonin. The deduced $K_d$ is indicated in parentheses. Averaged $K_d$s from two technical replicates (mean ± s.d.) are shown. (**b**) Proposed models of harmonin interactions. Top: Harmonin can exist in an autoinhibited state (autoinhibition) or can form a network (self-association) via its Cter (green star). Middle: SANS binding to the NPDZ1 domain of harmonin or harmonin-a binding to the Myo7 MF2 domain are incompatible with harmonin self-association. Bottom: Harmonin-b (orange) cannot self-associate via its Cter since its sequence differs. Instead, harmonin-b can participate into the network via its CC2 binding to NPDZ1.PDZ2. (**c**) Schematic diagrams summarizing the known interaction affinities within the tripartite complex involving harmonin isoforms. Left: Nearly identical interactions involving harmonin-a are found in microvilli and stereocilia, where ANKS4B and SANS can each bind to the MF1 of both Myo7 isoforms. Right: Interactions with harmonin-b in stereocilia will differ because its long Cter extension is not compatible with binding to MF2.

**Table 5 | Characterization of harmonin interactions.**

| Harmonin auto-inhibition/self-association | | $K_d$ |
|---|---|---|
| NPDZ1 | PDZ3c (isoA) | $10.0 \pm 3.0\,\mu M$ ($n = 3$)* |
| | $^{YFP}$PDZ3c (isoA) | $14.2\,\mu M$ |
| $^{YFP}$NPDZ1 | PDZ2-end (isoA) | $4.2\,\mu M$ |
| NPDZ1.PDZ2 | $^{YFP}$PDZ3c (isoA) | $3.2\,\mu M$ |
| $^{YFP}$NPDZ1.PDZ2 | PDZ3c (isoA) | $5.2\,\mu M$ |
| FL | $^{YFP}$PDZ3c (isoA) | $1.9\,\mu M$ |
| $^{YFP}$FL | PDZ3c (isoA) | $1.4\,\mu M$ |
| FL $\Delta$Cter | $^{YFP}$PDZ3c (isoA) | $2.3\,\mu M$ |
| $^{YFP}$NPDZ1.PDZ2 | PDZ3 $\Delta$Cter | n.b. |
| $^{YFP}$NPDZ1.PDZ2 | PDZ3c (isoA) F551V-F552V | n.b. |
| PDZ2-end (isoA) | $^{YFP}$PDZ3c (isoA) | n.b. |
| $^{YFP}$NPDZ1.PDZ2 | PDZ3c (isoB) | n.b. |
| | | |
| $^{YFP}$FL | NPDZ1 | $11.3\,\mu M$ |
| $^{YFP}$FL | NPDZ1.PDZ2 | $3.4 \pm 0.1\,\mu M$ ($n = 2$) |
| $^{YFP}$FL | PDZ3 $\Delta$Cter | n.b. |
| FL | $^{YFP}$FL $\Delta$Cter | $2.5\,\mu M$ |
| $^{YFP}$FL | PDZ3c (isoA) F551V-F552V | n.b. ($n = 2$) |
| $^{YFP}$FL | PDZ3c (isoA) D546R-F551V-F552V | n.b. |
| FL $\Delta$Cter | $^{YFP}$NPDZ1.PDZ2 | n.b. |
| $^{YFP}$FL $\Delta$Cter | PDZ2-end | $2.0\,\mu M$ |
| $^{YFP}$FL $\Delta$Cter | FL $\Delta$Cter | n.b. ($n = 2$) |
| $^{YFP}$FL $\Delta$Cter | PDZ3c F551V-F552V | n.b. |
| | | |
| PDZ3c (isoA) | $^{YFP}$SANS$_{CEN-PBM}$ + NPDZ1.PDZ2 | w.b. ($> 100\,\mu M$) ($n = 2$) |
| PDZ3c (isoA) | SANS$_{CEN-PBM}$ + $^{YFP}$NPDZ1.PDZ2 | w.b. ($> 100\,\mu M$) |
| PDZ3c (isoA) | SANS$_{CEN-PBM}$ + $^{YFP}$FL $\Delta$Cter | w.b. ($> 100\,\mu M$) |
| PDZ3c (isoA) | $^{YFP}$ANKS4b + NPDZ1.PDZ2 | w.b. ($> 100\,\mu M$) |
| PDZ3c (isoA) | ANKS4b + $^{YFP}$NPDZ1.PDZ2 | w.b. ($> 100\,\mu M$) |

n.b. and w.b. stand for no binding (no fit possible) and weak binding (the end of the curve is missing, and only a rough estimate of the $K_d$ can be provided), respectively.
*ITC result; $n$ = number of independent experiments.

PDZ3c and the PDZ2-end construct (Table 5), showing that PDZ1 is essential for the interaction. Altogether, these results suggest that the PDZ3c domain is likely positioned so that the PBM may bind to the PDZ1 domain, while PDZ2 contributes to the interaction. Although the measured affinities are not high, covalent linkage between these domains in FL-harmonin-a will drastically increase the apparent local concentration, favouring this interaction. Consistent with the existence of this potentially 'auto-inhibited' state of harmonin, small angle X-ray scattering (SAXS) studies showed that FL-harmonin adopts a shorter conformation (Rg = $38.42 \pm 1.75$ Å, Dmax = 136.0 Å) compatible with a 'folded-back' structure with Cter PBM bound to NPDZ1; while FL-harmonin-$\Delta$Cter is a more elongated molecule (Rg = $43.41 \pm 1.53$ Å, Dmax = 156.1 Å) (Supplementary Fig. 7b).

The ability of harmonin-a's Cter to bind to PDZ1 could lead to a stable, auto-inhibited molecule or mediate inter-molecular self-associations (Fig. 5b, top). The PDZ3c domain binds to both FL- and FL-harmonin-$\Delta$Cter with similar affinities ($\sim 2\,\mu M$) (Fig. 5a, Table 5), suggesting the harmonin opening/closing equilibrium is fast enough to allow association with PDZ3c. Importantly, the results demonstrate that self-association of harmonin molecules require the Cter motif and PDZ1 (Table 5). SANS$_{CEN-PBM}$ and ANKS4B$_{CEN-PBM}$ bind strongly to NPDZ1.PDZ2 (Table 4) and competition experiments show that this association abolishes PDZ3c binding (Table 5). Altogether, the data indicate a rather loose and dynamic auto-inhibited state of FL-harmonin-a that may form inter-molecular chains via Cter/PDZ1 interactions (Fig. 5b). This scaffold of harmonin-a would however be incompatible with Myo7 recruitment via either MF2 or MF1/SANS (Fig. 5b). Whether FL-harmonin-a indeed can self-interact to form chains of higher oligomers and whether this network is impeded by SANS remain to be determined.

The CC2 of harmonin-b can interact with the NPDZ1.PDZ2 of harmonin present in all isoforms[24,35] (Fig. 5b, bottom). Association of harmonin-b could promote formation of a network with F-actin attachment points via the ABD of harmonin-b (Fig. 1b). It would also leave the Myo7a MF2 domain available for interactions with other partners. Altogether, the data presented here extends our understanding of the interactions within the tripartite complex (Fig. 5c). They further highlight the potential mechanism of regulation by the presence and/or different concentrations of each member of the tripartite complex and how it may enable the assembly of diverse Myo7/adaptor/harmonin complexes.

## Discussion

Cadherin-based connections are essential for the stability and function of stereocilia and microvilli as well as for sensing external forces at the cell surface. The results presented here reveal nearly identical interactions between members of the Myo7/harmonin-a/SANS:ANKS4B tripartite complexes that link cadherins to the actin-rich core in stereocilia and microvilli. Both Myo7a and Myo7b MF2 domains bind to harmonin-a PDZ3c with similar affinities via a novel mode. SANS and ANKS4B both can bind to the MF1 domains of either Myo7a or Myo7b, consistent with the conserved sequences and structures of their binding sites. Binding to SANS/ANKS4B provides Myo7 a second, indirect link to harmonin, allowing the formation of a tightly integrated tripartite complex. Furthermore, the interactions of harmonin-a's Cter motif with either harmonin NPDZ1.PDZ2 or Myo7 MF2 may play a role in regulating tripartite complex assembly. Altogether, these results highlight the common global landscapes shared by the stereocilia and microvilli tripartite complexes.

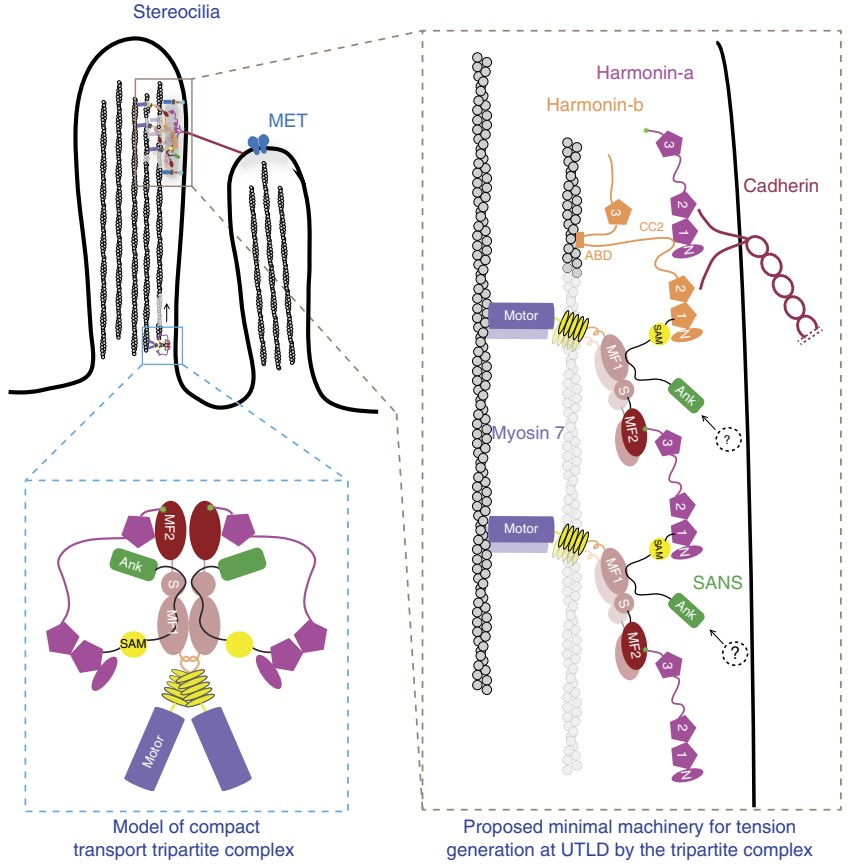

**Figure 6 | The proposed Myo7a tip link complex.** A cartoon view of stereocilia tip link. The upper tip link density (UTLD) is highlighted in grey. This region clusters several Usher syndrome proteins, as well as molecular motors implicated in adaptation such as Myo1c and Myo7a. A compact transport complex with Myo7a is shown at the base of the stereocilia moving up towards the tip link; and the proposed model shows that this complex consists of harmonin-a, SANS and Myo7a (blue box). Brown box: a proposed, much-simplified model showing a potential mechanism for Myo7a recruitment to the UTLD based on known interactions within the tripartite complex. Harmonin-a can crosslink two motors by interacting directly with a Myo7a via PDZ3c/MF2 interactions (mediated by the Cter motif, green star) and indirectly with another Myo7 via SANS/MF1. The motor in this case can be either monomeric or dimeric (shown in lighter colours; potentially promoted by SANS dimerization, see Fig. 3e). This network would be compatible with association between different harmonin isoforms. Harmonin-b (in orange) would provide the means to tether this network to actin filaments via its ABD while its CC2 region could extend the network by recruiting another harmonin. Thus, different numbers of Myo7a motors and harmonin-b could potentially be crosslinked and regulate the stiffness and extent of movement of this Usher protein assembly network, which would also bind the cadherin tails via harmonin. Also indicated is the potential interaction of the SANS ANK domain with other unknown partners (dotted circle). MET: mechano-electrical transduction channel.

Localization of tripartite complex members[14,19,23,35,40,41] indicates that Myo7 likely functions both as a transporter and as a tension generator at the apical link of stereocilia and microvilli. The structural, biochemical, and mutagenesis data suggest the modular nature of the complex and how it could specify different roles for Myo7. For driving long-range transport, the association of harmonin-a to two sites of the same Myo7 tail (Fig. 3e, intradimer and interdimer; Fig. 6, blue box) would strengthen a stable and tightly packed tripartite complex, while preventing harmonin-a from self-association via its Cter motif (Fig. 5b). This ternary complex would form a compact cargo with an active, likely dimerized Myo7 able to walk on bundled actin filaments to deliver specific partners[42]. It should be noted that while forced dimers of Myo7 can move along filopodia or microvilli[16,42], little is known about native Myo7 dimerization and how its motor function is activated in the context of the tripartite complex.

Clustering of several motors at the apical links of microvilli or stereocilia is likely required to efficiently apply tension and respond to external forces. Multiple harmonin isoforms are expressed in sensory hair cells[23,39], and they may form different networks of Usher proteins for clustering molecular motors and anchoring each type of link to the actin cytoskeleton. Stoichiometric cadherin/harmonin interaction has been proposed to cluster motors to the UTLD[43]. However, the number of cadherin tails present at the stereocilia tip link[10,44,45] is probably too low to cluster a sufficient number of motors[14]. A scaffold of harmonin-a formed via its N-ter/C-ter connections also cannot cluster motors to work as an assembly since this interaction is incompatible with direct recruitment of the Myo7 tail (Table 5, Fig. 5b).

A simple conceptual model for the potential role of harmonin-a in recruiting multiple Myo7a motors can be proposed (Fig. 6, brown box). The multidentate Myo7 tail binding sites in harmonin-a, together with its flexible linker region, could assemble tripartite complexes into a network connecting several Myo7 motors (see example in Fig. 3e, crosslinking by harmonin). While all of the binary interactions in this assembly are identical to those in the compact transport complex (Fig. 6, blue box), the flexibility within each protein allows the formation of distinct complexes. The harmonin CC region and/or linkers in SANS could allow the relative orientation

between each molecule to vary (Fig. 3d), potentially triggered by the increase in local concentration of the complex members and/or other signals present at the tip link, and thus promote 'clustering/crosslinking'. Self-association of harmonin-a would reduce the number of motors recruited; while SANS/ANKS4B binding to harmonin shifts the balance towards an organized scaffold that recruits molecular motors (Figs 5b and 6). The harmonin-b isoform, abundant at the stereocilia tip link, cannot bind either NPDZ1.PDZ2 or Myo7 MF2 via its extended Cter (Tables 1 and 5). Thus, clustering with harmonin-b would be based on its CC2-mediated interactions with NPDZ1.PDZ2 of other harmonins[35], as well as its binding to SANS; and would provide actin attachment points via its ABD. Changing the number of harmonin-b molecules participating in this network could impact the 'tightness' of the linkage between cadherins and the actin core, consistent with the phenotype of Ush1c[dfcr-2J/dfcr-2J] mutant mice, where loss of harmonin-b leads to an impairment in the extent and speed of adaptation after a stereocilia deflection[36,37]. Whether this simplest network or other types of assembly between harmonin and Myo7 molecules can form and would be adequate to apply force on cadherin tails requires further studies.

The molecular mechanism regulating tension/adaptation at the UTLD and the proteins involved are complex. The architecture of the Usher complex found at the stereocilia tip link needs to be tuned to contribute to the different sensitivities and selectivities from hair cell to hair cell specialized for different frequencies[46–48]. An important property of the model presented in Fig. 6 is its ability to be tuned for different stiffness. The amount and ratio of harmonin-a and -b isoforms would control the 'spring' of this motor assembly, the number of links connecting cadherins to the actin cytoskeleton, and the response it could have to deformation and force generation. The expression profile of different harmonin isoforms within stereocilia bundles as well as along the length of the cochlea is not well-characterized. However, a base-to-apex gradient of harmonin mRNA expression in the cochlea was observed[49], suggesting that changes in harmonin levels could indeed contribute to determining hair cell responses and sensitivities to different frequencies. The ability of harmonin-b to self-assemble and to bind F-actin may thus represent a delicate and refined mechanism for regulating the dynamics of the tip link protein assembly. Thus, variations in the assembly and the architecture of the tripartite complexes associated with each type of link and the resulting number of connections the network makes to the actin cytoskeleton are likely achieved by differential expression of harmonin isoforms. Other partners and motors could participate in this network to further fine-tune adaptation. One such motor is Myo1c (refs 50,51) that is broadly distributed in hair cells and concentrated in UTLD[51,52]. Whether and how Myo1c interacts with any member of the tripartite complex remains to be determined.

The interplay between Myo7 motors and various harmonin isoforms can result in distinct motor assemblies to fulfil specific developmental and mechanotransduction roles. Characterization of the force generated by motors and evaluation of the stiffness of different motor/harmonin assemblies is now needed to understand how those functions are accomplished. Isoform-specific functions as means of regulation has been observed for espin, whirlin and Myo15 (refs 53–55), with distinct isoforms trafficking selectively to different locations within stereocilia to independently modulate stereocilia elongation. Consequently, the development, maintenance and mechanotransduction of these cellular protrusions are regulated through a sophisticated, multi-layered mechanism that involves not only the function of the individual proteins and their

constituent isoforms but also their precise expression, regulation and localization profiles. How the interchange between each isoform can direct the assembly of different complexes and how these networks orchestrate the intricate functions of cell mechanotransduction remain to be elucidated.

*Note added in proof:* While this paper was under final review, Li et al.[70] reported the structure of the binary harmonin-a PDZ3c/Myo7b MF2 complex and presented complementary biochemical and cell biological results on the interactions between the Myo7b tripartite complex proteins.

## Methods

**Construct design.** The MF regions of the human Myo7a and Myo7b genes (UNIPROT Q13402-1; Q6PIF6, respectively) and harmonin-a (UNIPROT Q9Y6N9-1) were PCR cloned (Strataclone, Agilent or Gibson assembly, New England BioLabs) from available cDNA clones[3,5] (the Myo7A clone was a kind gift of Dr Lee Sweeney). Regions encoding either the Myo7a's or Myo7b's MF1.SH3 and MF2 (Fig. 1) were cloned into a modified pET-14 plasmid (Novagen/EMD Millipore) whose thrombin cleavage site was replaced by an SSG linker. Human harmonin-a and ANKS4B were PCR amplified from full-length cDNA clones[3] and cloned by TA cloning (Stratagene, Aglient). The full-length SANS gene and the 3′ end of the harmonin-b gene encoding the C-terminal extension were synthesized (Genscript and IDTDNA, respectively). The region of interest for each gene was ligated into pGST-parallel vector[56] or a modified pET14 or pET14-YFP expression plasmid using standard restriction cloning methods. The resulting pGST-parallel harmonin constructs all have an N-terminal GST tag followed by a Tobacco Etch Virus protease cleavage site. The ANKS4B and SANS constructs have an N-terminal 6xHis or 6xHis-YFP tag. Mutations or deletions were introduced either using the Quikchange Multi Lightning (Agilent) or Q5 Mutagenesis (New England Biolabs) systems per manufacturer's instructions. The sequences of primers and synthetic genes are listed in Supplementary Table 3.

**Protein expression and purification.** All 6xHis fusion proteins were expressed in either BL21-AI, BL21 (DE3) Rosetta (Invitrogen) or BL21-Gold (DE3) (Agilent) *Escherichia coli* at 20 °C after induction with 0.2 mM IPTG with or without 0.2% L-arabinose, accordingly. Selenomethionine derivatives were prepared using B834 (DE3) *E. coli* (Agilent) and Selenomethionine medium from Molecular Dimensions. The cells were collected by centrifugation, frozen in liquid N$_2$ and stored at − 80 °C. Frozen cells were thawed and lysed using a TS cell disruptor (Celld). The soluble fraction of the lysate was applied to a Histrap FF crude column (GE Healthcare) and the His fused proteins were eluted with 20 mM HEPES pH 7.5, 200 mM NaCl, 0.5 mM TCEP, 0.5 mM PMSF and 300 mM imidazole. The proteins were further purified by ion exchange chromatography on a HiScreen CaptoQ column (GE Healthcare) and gel filtration on a Superdex 200 or Superdex 75 column (GE Healthcare) in final buffer containing 20 mM HEPES pH 7.5, 100 mM NaCl, 0.5 mM TCEP, 0.5 mM PMSF. The final pool was concentrated, flash-frozen in liquid N$_2$ and stored in small aliquots at − 80 °C.

All harmonin constructs were expressed in BL21 (DE3) Rosetta (Invitrogen), as described above and purified using Glutathione Sepharose 4B affinity resin followed by size-exclusion chromatography. Samples submitted to ITC experiments or crystallization trials were digested with Tobacco Etch Virus protease and incubated with Glutathione Sepharose 4B resin. Small aliquots of purified harmonin constructs were flash-frozen in liquid N$_2$ and stored at − 80 °C. Mutants of MF or harmonin were purified similarly as wild type constructs.

**Crystallization and structure determination.** The Myo7b MF2 was crystallized using the hanging drop vapour diffusion method at 290 K by mixing 1:1 protein to reservoir solution (31–34% (v/v) PEG 400, 100 mM Tris–HCl pH 8.5 and 200 mM Li$_2$SO$_4$). The crystals were cryocooled in liquid nitrogen using the same reservoir solution. The selenomethionine derivative protein was crystallized in the same condition. The complex of Myo7b MF2 and harmonin-a PDZ3c was prepared by mixing the two proteins at 1:1.2 molar ratio and the sample was crystallized by mixing 1:1 protein to reservoir solution (7% (w/v) PEG 4000, 0.1 M HEPES pH 7.5, 50 mM MgCl$_2$) in hanging drops. The crystals were cryo-protected by sequential soaking in reservoir solution plus glycerol as a cryo-protectant and flash-frozen in liquid N$_2$. The complex of Myo7a MF2 and PDZ3c was prepared and crystallized similarly with reservoir solution containing 5% (w/v) PEG 4000, 0.1 M MES pH 6.5, 50 mM MgCl$_2$ and 10% (v/v) isopropanol. Crystals were optimized by microseeding and cryo-cooled in liquid N$_2$ in the reservoir solution plus 25% (v/v) glycerol.

X-ray data sets were collected at the Proxima-1 and Proxima-2 beamlines (SOLEIL synchrotron). The diffraction data sets were indexed and scaled with XDS[57]. Phases for the Myo7b MF2 structure were calculated by the single isomorphous replacement with the anomalous signal (SIRAS) method using native and selenomethionine derivative crystals collected at the peak wavelength of selenium. Seven Se sites were found by autoSHARP[58], and an initial model was

built by cycling between density modification and model building[59,60]. The model generated was refined with autoBUSTER (2.10.2; Global Phasing Ltd) and used as a search model for molecular replacement[61] using the native data. The model was subsequently edited and refined with Coot[62] and autoBUSTER. The Myo7b MF2-PDZ3c complex was solved by molecular replacement with Phaser[63] using the Myo7b MF2 structure as a search model. Model building and refinement were carried out with phenix.autobuild[64], Coot and phenix.refine[65]. The structure of Myo7a MF2-PDZ3c was then solved by molecular replacement using Myo7b MF2-PDZ3c as a search model. The model building and refinement were carried out similarly in phenix.autobuild, Coot and phenix.refine. Statistics on the data collection for all the structures and crystallographic statistics of the final models are summarized in Table 2. All structural figures were prepared using PyMOL (www.pymol.org/).

**Affinity measurements.** Isothermal titration calorimetry (ITC) measurements were performed on a Microcal iTC200 system (Malvern) at 10 °C. The protein samples were buffer-exchanged using PD-10 column (GE Healthcare) into 20 mM HEPES pH 7.0, 100 mM NaCl, 0.5 mM TCEP and 1 mM EGTA. Total of 3-µl aliquots of sample in the syringe were injected into the sample in the cell at 200 s intervals, enough time for the titration peak to return to the baseline. The titration data were analysed using Origin 7.0 (Microcal) using a single binding site model.

Microscale thermophoresis (MST) measurements were performed on a Monolith NT.115 (Nanotemper Technologies) using YFP-fusion proteins. Two-fold dilution series (16 in total) of the non-fluorescent protein were performed in the interaction buffer (20 mM Hepes pH 7.0, 100 mM NaCl (150 mM for FL-Harmonin-a), 1 mM EGTA, 0.5 mM TCEP and 0.05% (v/v) Tween 20). The YFP-fused partner was kept at a constant concentration of 100 nM. The samples were loaded into premium capillaries (Nanotemper Technologies) and heated for 30 s at 50% laser power. The affinity was quantified by analysing the change in thermophoresis as a function of the concentration of the titrated protein using the NTAnalysis software provided by the manufacturer.

For all affinity measurements, protein quality and monodispersity was assessed by Dynamic Light Scattering in interaction buffer (20 mM Hepes pH 7.0, 100 mM NaCl (150 mM for FL-Harmonin-a), 1 mM EGTA, 0.5 mM TCEP) with a DynaPro Plate Reader II (Wyatt Technology) at 20 °C. Only proteins and complexes displaying a single population with polydispersity < 25% were used.

**SEC-MALS.** Absolute molar masses of proteins were determined using size-exclusion chromatography combined with multi-angle light scattering (SEC-MALS). Protein samples (30.0 µl at ~20 mg ml$^{-1}$) were loaded onto a Superdex 10/300 Increase column (GE Healthcare) in 20 mM Tris pH 7.0, 100 mM NaCl, 1 mM EGTA and 0.5 mM TCEP at 0.5 ml min$^{-1}$ using a Dionex UltiMate 3000 HPLC system. The column output was fed into a DAWN HELEOS II MALS detector (Wyatt Technology). Data were collected and analysed using Astra X software (Wyatt Technology). Molecular masses were calculated across eluted protein peaks.

**Small angle X-ray scattering experiments.** SAXS data for the harmonin constructs were collected on the SWING beamline at the SOLEIL synchrotron ($\lambda = 1.033$ Å). 50 µl of FL-harmonin or FL ΔCter at ~2 mg ml$^{-1}$ were injected onto a size-exclusion column (SEC-3 300 Å Agilent), using an Agilent HPLC system[66] in 20 mM HEPES pH 7.0, 200 mM NaCl, 0.5 mM TCEP and 1 mM EGTA. Data reduction, frame averaging and buffer subtraction were done in FOXTROT, a dedicated home-sourced software, and the subsequent data processing and analysis were carried out with PRIMUS and other programs of the ATSAS suite[67]. The radius of gyration $R_g$ was derived by the Guinier approximation using PRIMUS. The program GNOM[68] was used to compute the pair-distance distribution functions, $p(r)$ and the maximum dimension of the macromolecule, $D_{max}$.

**Data availability.** The atomic coordinates and structure factors have been deposited in the Protein Data Bank, www.pdb.org, with accession numbers 5MV7 (Myo7b.MF2), 5MV8 (Myo7b.MF2/PDZ3c), 5MV9 (Myo7a.MF2/PDZ3c). The authors declare that all relevant data supporting the findings of this study are available on reasonable request.

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

## Acknowledgements

We thank beamline scientists of PX1, PX2 and SWING (SOLEIL synchrotron) for excellent support during data collection and G.Jousset for assistance with cloning. We thank Drs P.Martin, H.Lee Sweeney and N.Courtemanche for critical reading of the manuscript. We are grateful to Drs M.Tyska and H.Lee Sweeney for sharing cDNA clones. V.J.P.-H. is the recipient of a fourth year fellowship from Ligue Nationale Contre le Cancer. M.A.T. was supported by NSF Grant MCB-1244235 and the University of Minnesota Foundation; A.H. was supported by grants from the CNRS, ANR-13-BSV8-0019-01, FRM ING20140129255, Ligue Nationale Contre le Cancer and Association pour la Recherche sur le Cancer – Subvention Fixe. The AH team is part of Labex CelTisPhyBio 11-LBX-0038 and IDEX PSL (ANR-10-IDEX-0001-02-PSL).

## Author contributions

I.-M.Y. and V.J.P.-H. contributed equally to this work. I.-M.Y. and V.J.P.-H. determined X-ray structures and analysed them with A.H. V.J.P.-H. collected and analysed the SAXS data. Biochemical experiments were performed by Y.S. and M.A.T. with the help of H.S., I.-M.Y. and C.K. MST experiments were performed by D.M., V.J.P.-H. and H.S. ITC experiments were performed by D.S. A.H. conceived the project and oversaw the experiments. A.H. and M.A.T. analysed the results and wrote the manuscript with help from V.J.P.-H. and I.-M.Y.

## Additional information

**Competing interests:** The authors declare no competing financial interests.

