## [Peer review file · Nature Communications]

Reviewers' comments:

Reviewer #1 (Remarks to the Author):

The Yu et al manuscript from the Titus and Houdusse groups describe high-resolution structural studies of Myo7a and Myo7b and how they interact through their MyTh4/FERM domains with adaptor proteins, SANS (USH1G) and harmonin (USH1C). This macromolecular complex interacts with the F-actin cytoskeleton in hair cell stereocilia and each member of the complex is necessary for normal hearing and vision. Its disruption by mutations in MYO7A, SANS or USH1C are causes of Usher syndrome. Harmonin also binds cadherin 23, which is one of the two proteins that contributes to the tip-link, and is necessary for the mechanotransduction of sound.

The structural data provided in the authors' manuscript is convincing, sets a high standard and is timely and important. The manuscript is written in clear language, but in a few sections it lacks depth and scholarship for reasons mentioned below and can be significantly improved with additional effort.

Major points:

1. The results sections of the manuscript makes for heavy reading, and at times, I found myself losing track of where the story was heading. I would recommend that the authors try to scatter more discussion into the results section, so that the relevance of your findings are highlighted as you go along. It will make the data easier to digest and follow.
2. On page 3, the introductory comments to the molecular composition of stereocilia tip links (protocadherin-15 and cadherin-23) is correct, as far as it goes. However, what is missing about tip link molecular composition in the introduction suggests some naiveté. During development, tip links appear to be composed of protocadherin-15 on both the upper side and lower side of stereocilia. Moreover, after breakage of a mature tip link, which is likely to be a frequent occurrence, an early step in its regeneration to restore mechanosensitivity involves temporarily protocadherin-15 molecules on BOTH ends (Indzhukulian et al., 2013 Plos Biology). Don't over simplify tip link structure/composition.
3. Similarly, on page 3, beginning with line 79, the authors state that "Loss or mutation of any one of these proteins results in Usher syndrome type 1 (USH1)." Again, this is correct but overgeneralized. Numerous studies from many different research groups have demonstrated that mutations of the genes encoding myosin 7a, harmonin, cadherin 23 and protocadherin 15 are also associated just with nonsyndromic deafness DFNB2, DFNB12 and DFNB23. Briefly elaborating on this to help new readers who are not familiar with hereditary deafness.
4. In the introduction on page 4 and in the Discussion page 12 line 372, the authors touch upon the idea that myosin 7 can switch between being a transporter of cargo to a force sensor or anchor at the "apical link". How might this switch occur? Also, what is the evidence that Myo7a might be dimerized in vivo?
5. Line 443. This is a nice hypothesis. Many molecules expressed in the cochlea exhibit base (high frequency) to apex (low frequency) gradients. I wonder if there is any published data on harmonin that would help support your argument?
6. Line 452. The authors introduce Myo1c as involved in 'slow' adaptation. Up until this point in the manuscript, there has been no description of what adaptation is, let alone there are fast and slow components. Fast adaptation is thought to occur near to the site of the MET channel, on the lower end of the tip link. These concepts need to be introduced earlier in the Introduction, so that reader is aware of them. The origin of slow adaptation is controversial, and has been suggested to be Myo1c and Myo7a based at the upper end of the tip link, but these still remain working models,

and I think the authors should allude to this at some point.

7. In the manuscript in a few locations, Myo7a in stereocilia is either said to be or implied to be localized in stereocilia at the upper tip link density (UTLD). This statement is based on observation published in Grati et al., 2011 (authors' reference 7). In Grati et al., Myo7a is localized at the UTLD. However, in earlier papers from others (see Sentfen et al., 2006; Hasson et al 1997), Myo7a is localized along the length of stereocilia. Sentfen et al., state that "In agreement with previous findings, we observed that MYO7A was expressed more broadly throughout stereocilia (Hasson et al., 1997)." . Why are you confident that in the Grati and Kachar paper the localization of myosin 7a is so tightly localized just to the UTLD? What might account for this discrepancy between earlier observations and Grati and Kachar 2011? Perhaps the difference in localization in Grati and Kachar (2011) from earlier studies might be due to some epitopes being masked for certain antisera against myosin 7a but not for other antisera. Alternatively, the affinity of the antisera for myosin 7a may be very different. Maybe a "weak affinity" antisera to myosin 7a would show localization predominantly where there is a higher concentration of myosin 7a and this might explain the observations in Grati and Kachar 2011. There certainly are other possible explanations. Justify your confidence in the data on myosin 7a localization in Grati and Kachar 2011. But please don't rely on prettier images and cartoons as your defense.

8. The authors propose some elaborate models for how harmonin, SANS and Myo7a might assemble into larger macromolecular structures (Figure 5C). I think these are perfectly valid to speculate about. However, since you have not directly shown that these structures can, or do assemble, there needs to be a sentence somewhere stressing the limitation of the current data sets.

Minor points:

1. The authors inadvertently use two very different definitions for the term "isoform". Myosin 7a and myosin 7b are encoded by two different genes. Yet, myosin 7a and 7b are said to be isoforms (page 8, line 231 and page 9, line 260, for example). Despite the continued use of isoforms by biochemists to refer to distinct members of a family of proteins encoded by paralogous or orthologous genes, this is both confusing and incorrect. The concern is that in the manuscript the authors also refer to isoforms of harmonin (page 5, line 136). In this case, the authors are using the term "isoform" correctly. The different isoforms of harmonin arise from alternative splicing of a single gene. This confusion in the use of isoforms for distinctly different genes and, at the same time, for alternative splice isoforms of a single gene, must be corrected in the manuscript

2. Abstract. The sentence where you refer to high resolution structures – it is not clear if you are referring to your study or that of another group. Please clarify!

3. Abstract. Tip links are introduced in the final sentence, as if the reader is expected to know what they are. Please define these earlier in the abstract, otherwise the final sentence does not make sense to a non-specialist.

4. Introduction, first sentence. Mechanotransduction between cells is a broadly defined phenomena, it certainly is not limited to just cadherin based interactions.

5. Line 66. Are microvilli not there to increase the surface area for absorption also?

6. Line 86. The way this sentence reads to me, it implies that mutations in Myo7b/ANKS4B cause stereocilia disorganization. I don't think you mean that. Please clarify.

7. In several places in the manuscript you mentioned where you could not detect interactions published by other groups. Were there differences in buffer composition, or in assay methodology?

8. Line 146. Briefly explain what you mean by "induced fit".
9. Line 198. Does "strictly conserved" mean 100% identical? 100% similar? And for what species is it "strictly conserved"?
10. Line 268. (A1128E.R1129E.K1192E) Does this mean three amino acid substitutions in the same protein or any one of the substitutions will abrogate binding? Also, "(A1128E.R1129E.K1192E)" is confusing nomenclature.
11. Line 309. How do you know that "four mutations significantly weaken the interaction with PDZ3c" such that this interferes with the formation of strong stereocilia linkages,.."? Is this second part ("weaker stereocilia linkages") a prediction or was it experimentally demonstrated. Please clarify.
12. Line 322. There is an awful lot of data in figure 5A, and yet the authors do not really describe it in the main text. Is this table critical, then?
13. Line 355. I would argue that quite a lot is known about how forces are sensed in the auditory system. This sentence is not really adding anything to your discussion.
14. Line 375. "For driving long-range transport, the association of harmonin-a to two sites of the same Myo7 tail (Fig. 4Fiv, 6, blue box)". Do you mean Figure 3?
15. Line 436. Adaptation mechanisms are still not well defined. I would not assume they are necessarily sophisticated, we just don't know.
16. In figure 1, panel B, there is room to spell out identity and similarity (don't abbreviate). Is the sequence for the harmonin PDZ1, PDZ2 and PDZ3 for isoform 1a the same amino acid sequence as the PDZ domains 1, 2 and 3 for harmonin isoforms b2 and b3? If the answer is yes, the color coding could be the same for the sake of clarity. For the ANKS4B (and mentioned in the legend), what exactly are you calling linkers? Sequence between ANK and CEN and between CEN and SAM domains? Please label in the figure to clarify what is meant by "linker" in the legend.
17. For supplementary Figures 5 and 6, provide the accession numbers for the 8 and 14 proteins, respectively, that are aligned in these figures.
18. Figure 4A. Can you add in the wild-type binding affinity to this table? It will help make the contrast with your deafness associated mutants.
19. Supp Fig 4. Please add buffer conditions and temperature used for ITC experiments to the figure legend.

Reviewer #2 (Remarks to the Author):

This reviewer was asked to specifically assess the MST data and the authors provide extensive microscale thermophoresis data to characterize the intramolecular interaction of harmonin protein and intermolecular interactions between harmonin, myosin and SANS or ANKS4B proteins and domains thereof, using a plethora of single, double or triple point mutations as well as truncated proteins. Reporter proteins were specifically YFP-labelled target proteins. The MST method is well suited to investigate interactions between proteins and dependence of certain residues and domains of the interacting proteins.

Sufficient details are given about the MST experimental conditions and the characterization of the recombinant proteins investigated to enable evaluation of the presented data. The concentration of 100 nM of fluorescently labelled proteins is adequate for measurements of the Kds that are mostly in the low μM range.

One could argue on the significance of presented Kd values $>50 \mu\text{M}$. All MST binding curves shown in the manuscript or in the supplemental data were performed with highest concentrations of the ligand/interaction partner of $\sim 100 \mu\text{M}$. As a result, for some of the curves presented (e.g. PDZ3c vs NPDZ1 in F5B or Myo7bMF2 vs PDZ3c F551V+F552V in SF4D) and likely others (that are not presented in the manuscript), saturation of the fully bound state is not reached and thus presented Kd values are overestimated or represent rough estimates. For some interactions (e.g. in table 1 or all in F4A) this is clearly stated and binding is characterized as w.b. (weak binding, with MST curved not reaching a plateau) and Kds are given estimates ($>x \mu\text{M}$).

This should be done for all Kd $>50 \mu\text{M}$, as there are a few interactions, where exact Kds are presented as (e.g. $602 \pm 30 \mu\text{M}$ for M7b MF2 L2083W vs yfp-PDZ3c; Table 1).

Many experiments were obviously only performed once, though where curves are presented, they seem reliable and justify the conclusions drawn from the resulting Kd values and support the binding modes proposed in the manuscript.

Reviewer #3 (Remarks to the Author):

This paper describes the structure and assembly of the Myo7a and Myo7b MyTH4-FERM domains with the C-terminal domain of harmonin. The work overall is of high standard and leads to a much clearer impression of how Myo7a/SANS/Harmonin and Myo7b/ANKS4B/Harmonin complexes form in microvilli. The quality of the presentation is lacking for work of such high caliber. First and foremost, in its present form, it will be exceedingly difficult for anyone outside of a few experts to follow the story presented in the manuscript. Somehow the authors need to do a better job of conveying the importance of their findings and how they differ from or enlighten the earlier studies. The figures need to be improved. A few detailed comments are listed below

The first page of the introduction is excellent, but shortly thereafter it degenerates into Alphabet soup. This might be common in the field (looking back at the previous papers), but it does not make the literature accessible to the general reader.

Page 4 of the introduction. It would be very helpful to the non-specialist to describe up-front what structures are already available for the Myo7 tail domains. This would help place the current studies in context. There is an implication in the text of earlier structural work, but the reader has to dig out what is known. A direct statement is essential.

Line 98. The acronym PBM should be defined here and not later (line 161)

Line 143. Are three decimal places justified for the r.m.s. deviations? The authors should read the manuscript carefully for their claims of significance and adjust accordingly.

Page 5 and onwards. It would be helpful to include a description of the buried surface area for the interactions between the MyTH4-FERM domains and the Harmonin domain. This should include both the Cter and the PDZ domains (separately). This should describe both the total area and the hydrophobic and polar contributions. This would place the structures in context and would allow the general reader to gain an understanding of the nature of the interactions.

Line 248 and onwards. The authors need to explain the division of the CEN domain into two regions CEN1 and CEN2 otherwise it is not clear where these regions, divisions, or perhaps domains come from. This could be defined in Figure 1B with a change to the cartoon and caption.

Line 318. Although the interaction between NPDZ1 and PDZ3c is weak (it should be weak!) its effect will be considerably higher because of the covalent linkage between the domains which will dramatically increase the apparent concentration.

Line 352 The discussion covers some very interesting territory, but it is too long (almost four pages of text). There is a lot of speculation associated with this text that could be truncated without loss (wherever the word "likely" occurs some text can be removed).

Figure 2A. the electron density adds nothing but noise to this figure and should be removed. Much better density is shown in the supplementary material.

Figure 2B. The authors should decide what is important to show in this figure and downplay the rest. I believe it is the Cter that is important, but it is lost in the details of the MyTH4 and FERM domains. Not-fancy helices might help in Pymol. You might also consider toning down the colors and reducing the width of the ribbons.

Figure 2C. It is very hard to see the small differences in this figure with the choice of colors and widths of secondary structural elements.

Figure 1D. This might belong in the supplementary material. The actual information content relative to the discussion is quite small. It is a very difficult figure to follow. The text is very small.

Also the distances should not be displayed to two decimal places. The panel is very noisy and will not print well. Ideally the authors should edit the figure and remove the redundant information and only include that which is meaningful to the paper.

Figure 3. There is too much information in this figure. Again the division of CEN into CEN1 and CEN2 based on sequence and structural studies should have been explained earlier. A non-expert might ask why CEN2 is restricted to only a short segment of the highly conserved region shown here. Panels B, C, E, and F are too small. The text in Panel C is completely illegible. Panel D is really a table not a figure. Panels E and F are really important since they define the model arising from this work. They are far too small. Panel F could use some more explanation (labels) within the figure to be truly useful.

Figure 4. The interesting visual information in this figure is the location of the mutations in the interface and on the surface. This is lost in the current rendition because the colors from the ribbon dominate the visual experience. The authors should downplay the ribbons and Crayola colors and attempt to bring out the positions of mutations. The authors might consider using a level of transparency for the ribbons or muted colors.

Figure 5. There is a lot of great information in this figure, but the text is too small in Panel B and the labels in Panels C and D are also tiny in places.

Figure 6 contains a lot of speculation that is beyond the scope of the current paper.

Detailed Response to Reviewers:

To the Reviewers:

We thank you all for your careful consideration of our manuscript and positive evaluation of the work. Your constructive comments were quite helpful and in the course of addressing them we believe that the manuscript is much improved. The revision specifically focused on the comments from Reviewer #1 on the clarity and depth of our introduction and discussion and those from Reviewer #3 on suggestions regarding our presentation. We also took care to address the comments from Reviewer #2 with regards to the presentation of the binding data throughout the manuscript. Please note that the title of the manuscript had to be changed to conform with the Nature Communications “no punctuation” formatting requirements.

The point-to-point response to the reviewers’ comments is provided below, with the reviewer’s comments in black and our replies in blue. The numbering of pages, figures, and references, where applicable, applies to the **revised** version of the manuscript. Specific changes in the text that address a reviewer’s comment(s) are highlighted in yellow. Note that there were substantial changes made to the Results section in terms of writing and organization so as to make our findings clearer, per Reviewer #1, and more accessible to the general reader, per Reviewer #3. Where the text was significantly altered we will note that for the reviewer and provide a general explanation of how we incorporated their suggestions into the writing.

Point-to-point response to the comments of reviewers:

Reviewer #1 (Remarks to the Author):

The Yu et al manuscript from the Titus and Houdusse groups describe high-resolution structural studies of Myo7a and Myo7b and how they interact through their MyTh4/FERM domains with adaptor proteins, SANS (USH1G) and harmonin (USH1C). This macromolecular complex interacts with the F-actin cytoskeleton in hair cell stereocilia and each member of the complex is necessary for normal hearing and vision. Its disruption by mutations in MYO7A, SANS or USH1C are causes of Usher syndrome. Harmonin also binds cadherin 23, which is one of the two proteins that contributes to the tip-link, and is necessary for the mechanotransduction of sound.

The structural data provided in the authors’ manuscript is convincing, sets a high standard and is timely and important. The manuscript is written in clear language, but in a few sections it lacks depth and scholarship for reasons mentioned below and can be significantly improved with additional effort.

Thank you for your appreciation of the work and the insightful and constructive comments. They were quite helpful and the manuscript has been strengthened due to the introduction of the suggested changes/clarifications. The specific changes that were made to the manuscript are described below.

Major points:

1. The results sections of the manuscript makes for heavy reading, and at times, I found myself losing track of where the story was heading. I would recommend that the authors try to scatter

more discussion into the results section, so that the relevance of your findings are highlighted as you go along. It will make the data easier to digest and follow.

We agree with the reviewer and have significantly revised the Results section to include both a summary of the significance/relevance of a given series of experiments at the end of a section and also provide a better overview and rationale for them at the beginning. The description of results were also condensed and simplified in several places to get to the point in a clearer manner.

Substantial changes in the presentation were made all throughout so individual changes have not been highlighted.

2. On page 3, the introductory comments to the molecular composition of stereocilia tip links (protocadherin-15 and cadherin-23) is correct, as far as it goes. However, what is missing about tip link molecular composition in the introduction suggests some naiveté. During development, tip links appear to be composed of protocadherin-15 on both the upper side and lower side of stereocilia. Moreover, after breakage of a mature tip link, which is likely to be a frequent occurrence, an early step in its regeneration to restore mechanosensitivity involves temporarily protocadherin-15 molecules on BOTH ends (Indzhukulian et al., 2013 Plos Biology). Don't over simplify tip link structure/composition.

We thank the reviewer for pointing this out and emphasizing the importance of carefully describing the tip link composition. In our efforts to simplify the system for general readers we clearly overgeneralized. The Introduction now includes a brief description of the stereocilia links and the developmental progression. Due to space limitations we omitted mention of the fact that regeneration following damage involves recapitulation of the developmental program. **see new paragraph on page 3:**

“Stereocilia of increasing height are organized into bundles that are linked together by a variety of connections that undergo dynamic changes during development^{6,7}. These links serve to both maintain bundle morphology and enable force transmission. When stimulated, hair cell bundles are deflected and the MET channels are open, resulting in Ca^{2+} influx. They then undergo adaptation, a process that involves myosin motors, to reduce the electrical response of the hair bundle, thus preventing saturation and ensuring that the bundle remains sensitive to further stimuli^{8,9}. Tip links connect the top of a lower stereocilia, where the MET channel is localised, to the side of an adjacent, upper one; and are thought to contribute to resting tension and regulate adaptation. They appear early in development, initially composed of protocadherin 15 (PCDH15) dimers, then mature into a heterodimer of PCDH15 and Cadherin 23 (CDH23) homodimers localised to the lower- and upper-tip link density (LTLD and UTLD) in adjacent stereocilia, respectively^{8,10}.”

3. Similarly, on page 3, beginning with line 79, the authors state that “Loss or mutation of any one of these proteins results in Usher syndrome type 1 (USH1).” Again, this is correct but overgeneralized. Numerous studies from many different research groups have demonstrated that mutations of the genes encoding myosin 7a, harmonin, cadherin 23 and protocadherin 15 are also associated just with nonsyndromic deafness DFNB2, DFNB12 and DFNB23. Briefly elaborating on this to help new readers who are not familiar with hereditary deafness.

The reviewer has raised another excellent point and we have also included mention of the fact that nonsyndromic deafness can arise from mutations in several key components of the Usher complex (see pg 4).

“Several mutations in these proteins and/or cadherin are found to be associated with non-syndromic deafness DFNB and DFNA⁷, while others result in Usher syndrome type I (USH1), the most severe form of deaf-blindness characterized by profound congenital hearing loss and a prepubertal onset of retinis pigmentosa⁴.”

4. In the introduction on page 4 and in the Discussion page 12 line 372, the authors touch upon the idea that myosin 7 can switch between being a transporter of cargo to a force sensor or anchor at the “apical link”. How might this switch occur? Also, what is the evidence that Myo7a might be dimerized in vivo?

The reviewer raises a good point here and we are quite interested in knowing how the motor can switch between being a transporter or an anchor in a partner-dependent manner. This is unknown at present. The Discussion now includes some indication of how we envision this might happen but we refrained from going into too much detail at this point. We also added a sentence indicating that it is indeed unknown whether Myo7 is dimerized in vivo. (the text below was added to page 13)

“For driving long-range transport, the association of harmonin-a to two sites of the same Myo7 tail (Fig. 3e, bottom; 6, blue box) would strengthen a stable and tightly packed tripartite complex while preventing harmonin-a from self-association via its Cter motif (Fig. 5b). This ternary complex would form a compact cargo with an active, likely dimerised Myo7 able to walk on bundled actin filaments to deliver specific partners⁴³. It should be noted that while forced dimers of Myo7 can move along filopodia or microvilli^{16,43}, little is known about native Myo7 dimerisation and how its motor function is activated in the context of the tripartite complex.”

5. Line 443. This is a nice hypothesis. Many molecules expressed in the cochlea exhibit base (high frequency) to apex (low frequency) gradients. I wonder if there is any published data on harmonin that would help support your argument?

Yoshimura et al (2014, PLoS ONE) have shown what appear to be modest changes in harmonin mRNA expression between the apex and base regions of the cochlea (1.5- to 2-fold depending on the analysis method). While these mRNA expression data are indeed consistent with our speculative model, there does not appear to be any published data to our knowledge concerning harmonin protein levels, and particularly no information on an expression gradient of different harmonin isoforms are available

Nevertheless, the Yoshimura et al paper does suggest a functional role of harmonin, as well as a series of other proteins, in determining the tonotopy of cochlear hair cells. We thus now refer to this work in the text (see page 14).

“The expression profile of different harmonin isoforms within stereocilia bundles as well as along the length of the cochlea is not well characterized. However, a base-to-apex gradient of harmonin mRNA expression in the cochlea was observed⁵⁰, suggesting that changes in harmonin

levels could indeed contribute to determining hair cell responses and sensitivities to different frequencies.”

6. Line 452. The authors introduce Myo1c as involved in ‘slow’ adaptation. Up until this point in the manuscript, there has been no description of what adaptation is, let alone there are fast and slow components. Fast adaptation is thought to occur near to the site of the MET channel, on the lower end of the tip link. These concepts need to be introduced earlier in the Introduction, so that reader is aware of them. The origin of slow adaptation is controversial, and has been suggested to be Myo1c and Myo7a based at the upper end of the tip link, but these still remain working models, and I think the authors should allude to this at some point.

The reviewer is correct in pointing out that adaptation was neither properly introduced nor explained, in addition to the fact that much remains to be known about this process. We have now briefly introduced adaptation in the Introduction (see page 3) but refrained from going into too much detail due to the complexity of the process and the fact that our proposed models do not specifically address the mechanism.

“When stimulated, hair cell bundles are deflected and the MET channels are open, resulting in Ca^{2+} influx. They then undergo adaptation, a process that involves myosin motors, to reduce the electrical response of the hair bundle, thus preventing saturation and ensuring that the bundle remains sensitive to further stimuli^{8,9}. ”

A paragraph in the Discussion that touches on adaptation was also modified (see page 14).

“The molecular mechanism regulating tension/adaptation at the UTLD and the proteins involved are complex. The architecture of the Usher complex found at the stereocilia tip link needs to be tuned to contribute to the different sensitivities and selectivities from hair cell to hair cell specialized for different frequencies⁴⁷⁻⁴⁹. An important property of the model presented in Fig. 6 is its ability to be tuned for different stiffness. The amount and ratio of harmonin-a and -b isoforms would control the “spring” of this motor assembly, the number of links connecting cadherins to the actin cytoskeleton, and the response it could have to deformation and force generation. The expression profile of different harmonin isoforms within stereocilia bundles as well as along the length of the cochlea is not well characterized. However, a base-to-apex gradient of harmonin mRNA expression in the cochlea was observed⁵⁰, suggesting that changes in harmonin levels could indeed contribute to determining hair cell responses and sensitivities to different frequencies. The ability of harmonin-b to self-assemble and to bind F-actin may thus represent a delicate and refined mechanism for regulating the dynamics of the tip link protein assembly. Thus, variations in the assembly and the architecture of the tripartite complexes associated with each type of link and the resulting number of connections the network makes to the actin cytoskeleton are likely achieved by differential expression of harmonin isoforms. Other partners and motors could participate in this network to further fine-tune adaptation. One such motor is Myo1c^{51,52} that is broadly distributed in hair cells and concentrated in UTLD^{52,53}. Whether and how Myo1c interacts with any member of the tripartite complex remains to be determined “

7. In the manuscript in a few locations, Myo7a in stereocilia is either said to be or implied to be

localized in stereocilia at the upper tip link density (UTLD). This statement is based on observation published in Grati et al., 2011 (authors' reference 7). In Grati et al., Myo7a is localized at the UTLD. However, in earlier papers from others (see Sentfen et al., 2006; Hasson et al 1997), Myo7a is localized along the length of stereocilia. Sentfen et al., state that "In agreement with previous findings, we observed that MYO7A was expressed more broadly throughout stereocilia (Hasson et al., 1997)." . Why are you confident that in the Grati and Kachar paper the localization of myosin 7a is so tightly localized just to the UTLD? What might account for this discrepancy between earlier observations and Grati and Kachar 2011? Perhaps the difference in localization in Grati and Kachar (2011) from earlier studies might be due to some epitopes being masked for certain antisera against myosin 7a but not for other antisera. Alternatively, the affinity of the antisera for myosin 7a may be very different. Maybe a "weak affinity" antisera to myosin 7a would show localization predominantly where there is a higher concentration of myosin 7a and this might explain the observations in Grati and Kachar 2011. There certainly are other possible explanations. Justify your confidence in the data on myosin 7a localization in Grati and Kachar 2011. But please don't rely on prettier images and cartoons as your defense.

We thank the reviewer for raising this critical point. Indeed, we have been a bit uncertain about the difference in striking and seemingly exclusive staining for Myo7a at the UTLD shown in Grati et al (2011) and the more widely distributed (along the length of the stereocilia) localizations shown by other groups. We have now de-emphasized the focus on the localization of Myo7a to the UTLD and taken care to indicate that Myo7a is found along the length of stereocilia (see page 3).

"While Myo7a and harmonin are found to localise along the length of stereocilia, all three Usher proteins are also concentrated at UTLD where they associate with the tip link and regulate the function of MET¹³⁻¹⁵ "

8. The authors propose some elaborate models for how harmonin, SANS and Myo7a might assemble into larger macromolecular structures (Figure 5C). I think these are perfectly valid to speculate about. However, since you have not directly shown that these structures can, or do assemble, there needs to be a sentence somewhere stressing the limitation of the current data sets.

The reviewer is correct to point out that we need to alert the reader to limitations of the available data. We have indicated some of the pieces of information that are lacking at present at the end of the Results and in Discussion.

"Whether FL-harmonin-a indeed can self-interact to form chains of higher oligomers and whether this network is impeded by SANS remain to be determined. " [Results, see page 12]

"Whether this simplest network or other types of assembly between harmonin and Myo7 molecules can form and would be adequate to apply force on cadherin tails requires further studies. " [Discussion, see page 14]

“Characterization of the force generated by motors and evaluation of the stiffness of different motor/harmonin complex assemblies is now needed to understand how those functions are accomplished.” [Discussion, see page 15]

Minor points:

1. The authors inadvertently use two very different definitions for the term “isoform”. Myosin 7a and myosin 7b are encoded by two different genes. Yet, myosin 7a and 7b are said to be isoforms (page 8, line 231 and page 9, line 260, for example). Despite the continued use of isoforms by biochemists to refer to distinct members of a family of proteins encoded by paralogous or orthologous genes, this is both confusing and incorrect. The concern is that in the manuscript the authors also refer to isoforms of harmonin (page 5, line 136). In this case, the authors are using the term “isoform” correctly. The different isoforms of harmonin arise from alternative splicing of a single gene. This confusion in the use of isoforms for distinctly different genes and, at the same time, for alternative splice isoforms of a single gene, must be corrected in the manuscript.

We are grateful to the reviewer for pointing out this confusing and inappropriate use of the term “isoform”. We have now changed “isoform” to “paralog” when describing the two different Myosin 7s.

2. Abstract. The sentence where you refer to high resolution structures – it is not clear if you are referring to your study or that of another group. Please clarify!

We changed the sentence to now read: “ We determine high-resolution structures...”

3. Abstract. Tip links are introduced in the final sentence, as if the reader is expected to know what they are. Please define these earlier in the abstract, otherwise the final sentence does not make sense to a non-specialist.

The limited space in the Abstract forced us to omit a definition of the tip link but we agree with the point raised by the reviewer that we should not expect a reader to know what it is. Thus we have removed “tip link” from the last sentence and simply say “... how switching between different harmonin isoforms can regulate the formation of networks with Myo7a motors and coordinate force sensing in stereocilia.” Tip link is now defined in Introduction.

4. Introduction, first sentence. Mechanotransduction between cells is a broadly defined phenomena, it certainly is not limited to just cadherin based interactions.

This is an excellent point - the first sentence that presents linkages between cells, stereocilia and microvilli in the introduction has been rewritten (see pg 3):

““Mechanotransduction is a process by which cells convert mechanical stimulus into electrochemical signals. Special classes of cellular protrusions, such as the stereocilia of sensory hair cells and microvilli on intestinal epithelial cells, are linked together by cadherin, enabling a coordinated response to extracellular stimuli.”

5. Line 66. Are microvilli not there to increase the surface area for absorption also?

Yes, this is a good point. We have now changed the sentence to indicate their role in increasing surface area (see pg 3).

“..Similarly, microvilli increase the surface area of epithelial cells for absorption and are tightly connected at their tips to provide an integrated barrier against resident gut bacteria.³.”

6. Line 86. The way this sentence reads to me, it implies that mutations in Myo7b/ANKS4B cause stereocilia disorganization. I don't think you mean that. Please clarify.

The reviewer is correct and now the paragraph discussing mutations in Myo7 complex members and the impact on microvilli organization has been incorporated into a revised paragraph in the Introduction (see pg 4). The final sentence of that paragraph now reads:

“Mutations in, or deletions of, any component of this tripartite complex results in disorganization of apical microvilli^{3,5,16}.”

7. In several places in the manuscript you mentioned where you could not detect interactions published by other groups. Were there differences in buffer composition, or in assay methodology?

The reviewer makes a good point - we should have indicated differences in assay methods used to measure (or not) the interactions studied here where there were discrepancies.

In brief, a number of early studies were carried out using yeast two-hybrid screens (e.g. Boëda, 2002; Siemens, 2002) alone or in combination with ectopic expression in cultured cells followed by IP. Later measurements with purified domains were carried out using buffer conditions close to ours (i.e. 100 mM salt). Where significant differences in interactions were observed we now point out whether or not similar assay methods/buffers were used in previous work

“This is in contrast to initial reports that did not detect a significant interaction²², despite the fact that the binding assays were performed at the same ionic strength (100 mM NaCl).” [pg 7]

“Thus, PDZ3 contributes modestly to the interaction with MF2. The Myo7a MF2 domain has been shown by pull-down experiment to bind to the NPDZ1.PDZ2 domains of harmonin²³...” [pg 8]

“Although the different techniques and/or conditions used for those binding assays might lead to the conflicting outcomes,” [pg 8]

8. Line 146. Briefly explain what you mean by “induced fit”.

The intention was to explain that binding of the PDZ3c to the MF domain did not alter its structure drastically but minor differences in the Cter conformation observed between Myo7a and Myo7b suggests the groove can have small-degree pliability to “fit” different partners. The

main point, however, is to illustrate that the conformation of the groove defines partner specificity. This description has been changed to make the meaning clearer for the reader (see pg 5).

“Limited structural changes are found in the Myo7b MF2 supramodule upon PDZ3c binding (Fig. 2c), with an overall r.m.s.d. of 0.63 Å for 414 Cα atoms between the bound and free structures. This limited conformational pliability upon cargo binding indicates that the groove of a particular FERM domain defines its cargo recognition. “

9. Line 198. Does “strictly conserved” mean 100% identical? 100% similar? And for what species is it “strictly conserved”?

We meant identical except Tyr2026^{Myo7a}/Phe1923^{Myo7b}. We cited Fig-S6 and changed the sentence to make the meaning clear (see pg 7).

“All of the residues present in the central FERM groove of Myo7b MF2 involved in binding to PDZ3c are conserved in both vertebrate paralogs, except for a small Tyr2026^{Myo7a}/Phe1923^{Myo7b} difference (Supplementary Fig. 6, arrows and #)”

10. Line 268. (A1128E.R1129E.K1192E) Does this mean three amino acid substitutions in the same protein or any one of the substitutions will abrogate binding? Also, “(A1128E.R1129E.K1192E)” is confusing nomenclature.

It refers to all three amino acids being substituted. We changed this to A1128E-R1129E-K1192E to be consistent with the annotation of all of our mutations (see pg 9).

11. Line 309. How do you know that “four mutations significantly weaken the interaction with PDZ3c” such that this interferes with the formation of strong stereocilia linkages,..”? Is this second part (“weaker stereocilia linkages”) a prediction or was it experimentally demonstrated. Please clarify.

This was a prediction so we have revised the text so as not to overstate the implication of the results (see pg 11).

“Interestingly, these four mutations significantly weaken the interaction with PDZ3c (Fig. 4), suggesting the importance of the Myo7a MF2/harmonin PDZ3c interactions in hearing and a functional role for harmonin-a.”

12. Line 322. There is an awful lot of data in figure 5A, and yet the authors do not really describe it in the main text. Is this table critical, then?

The table is indeed critical as it contains a good deal of supporting binding data describing the nature of the interactions between harmonin subdomains. Unfortunately, the significance of these data was lost in the way that the findings were presented. The Results section describing these data has been extensively revised to better convey these binding data (“Harmonin autoinhibition and Self Association section in Results - pgs 11 & 12) and the actual binding data now moved to new Table 4.

13. Line 355. I would argue that quite a lot is known about how forces are sensed in the auditory system. This sentence is not really adding anything to your discussion.

We removed this sentence, as it is true that it does not add to our discussion.

14. Line 375. "For driving long-range transport, the association of harmonin-a to two sites of the same Myo7 tail (Fig. 4Fiv, 6, blue box)". Do you mean Figure 3?

Yes! This was our mistake - it has been corrected and the panels in that figure were re-formatted to meet Nature Communication style requirement (new Fig 3e).

15. Line 436. Adaptation mechanisms are still not well defined. I would not assume they are necessarily sophisticated, we just don't know.

Thank you for pointing this out. It is quite fair to say that we do not know the precise mechanism of adaptation. However, the vast and diverse proteins present in stereocilia and their dynamic/differential distribution along the cell during development as well as at matured stage do suggest to us that there is likely a complex regulatory mechanism at play during adaptation but it is entirely fair to say that this not known at present.

The sentence has been revised and we now simply point out that adaptation is "complex". (see pg 14)

"The molecular mechanism regulating tension/adaptation at the UTLD and the proteins involved are complex"

16. In figure 1, panel B, there is room to spell out identity and similarity (don't abbreviate).

We have spelled out similarity and identity.

-- Is the sequence for the harmonin PDZ1, PDZ2 and PDZ3 for isoform 1a the same amino acid sequence as the PDZ domains 1, 2 and 3 for harmonin isoforms b2 and b3? If the answer is yes, the color coding could be the same for the sake of clarity.

The sequences of all three harmonin PDZ domains are identical in each of the harmonin isoforms.

However, for clarity in the model figures (Fig 3e and Fig 5b, c) and because of the important difference in the role of harmonin-b in dictating the function of the tripartite complex, we find it important to highlight the two with different colors.

A sentence has been added to the legend for Fig 1b to clarify this to be sure that the readers know that the PDZ domains are identical among harmonin isoforms.

"Although harmonin isoforms contain identical sequences in its N-ter, PDZ1, 2, and 3, and the CC1 domains, in order to highlight the different functional impact between harmonin-a and -b we represent them in different colors."

-- For the ANKS4B (and mentioned in the legend), what exactly are you calling linkers? Sequence between ANK and CEN and between CEN and SAM domains? Please label in the figure to clarify what is meant by "linker" in the legend.

The sentence below has been added to the legend to clarify this point and Fig 3a has been revised to clearly indicate the linker sequence.

"The CEN domain of ANKS4B/SANS is comprised of two highly conserved CEN regions (labelled 1 and 2) connected by a short linker (see also Fig. 3a)."

17. For supplementary Figures 5 and 6, provide the accession numbers for the 8 and 14 proteins, respectively, that are aligned in these figures.

The accession numbers are now provided in the legends.

18. Figure 4A. Can you add in the wild-type binding affinity to this table? It will help make the contrast with your deafness associated mutants.

In response to your earlier comment, and because of Reviewer-2's comments and to comply with Nature Communications figure requirements, we removed the table presented in Fig 4A and instead now present the original MST binding curves specify that all mutants have weak affinity in the legend.

19. Supp Fig 4. Please add buffer conditions and temperature used for ITC experiments to the figure legend.

The buffer and condition for ITC are detailed in the methods. We now also specify this in the legend.

Reviewer #2 (Remarks to the Author):

This reviewer was asked to specifically assess the MST data and the authors provide extensive microscale thermophoresis data to characterize the intramolecular interaction of harmonin protein and intermolecular interactions between harmonin, myosin and SANS or ANKS4B proteins and domains thereof, using a plethora of single, double or triple point mutations as well as truncated proteins. Reporter proteins were specifically YFP-labelled target proteins. The MST method is well suited to investigate interactions between proteins and dependence of certain residues and domains of the interacting proteins.

Sufficient details are given about the MST experimental conditions and the characterization of the recombinant proteins investigated to enable evaluation of the presented data. The concentration of 100 nM of fluorescently labelled proteins is adequate for measurements of the K_ds that are mostly in the low μM range.

One could argue on the significance of presented K_d values >50 μM. All MST binding curves shown in the manuscript or in the supplemental data were performed with highest concentrations of the ligand/interaction partner of ~100 μM. As a result, for some of the curves

presented (e.g. PDZ3c vs NPDZ1 in F5B or Myo7bMF2 vs PDZ3c F551V+F552V in SF4D) and likely others (that are not presented in the manuscript), saturation of the fully bound state is not reached and thus presented K_d values are overestimated or represent rough estimates. For some interactions (e.g. in table 1 or all in F4A) this is clearly stated and binding is characterized as w.b. (weak binding, with MST curved not reaching a plateau) and K_d s are given estimates ($>x \mu\text{M}$).

This should be done for all $K_d > 50 \mu\text{M}$, as there are a few interactions, where exact K_d s are presented as (e.g. $602 \pm 30 \mu\text{M}$ for M7b MF2 L2083W vs yfp-PDZ3c; Table 1).

The reviewer is correct to point this out and we thank them for this comment.

Per the reviewer's suggestion, in these cases we have now indicated weak binding (w. b.) whenever the measured K_d is weak and the MST curve is not reaching a plateau.

Some of those assays, however, were indeed performed with a very high concentration of protein and therefore a rough K_d can be calculated. To clearly indicate that even using such concentrated protein the fully bound state was not reached, we reported the rough K_d calculated by the software, while stating that it is not fully reliable (w.b.)

Many experiments were obviously only performed once, though where curves are presented, they seem reliable and justify the conclusions drawn from the resulting K_d values and support the binding modes proposed in the manuscript.

We would also point out that in those cases where a single measurement was taken, there were single measurements made for either the related proteins/mutants and in a few cases the same MST measurement was done for the same pair of proteins with the YFP switched (see Table 4 - FL-harmonin with PDZ3) or with a different method (both ITC and MST) that are consistent with each other.

Reviewer #3 (Remarks to the Author):

This paper describes the structure and assembly of the Myo7a and Myo7b MyTH4-FERM domains with the C-terminal domain of harmonin. The work overall is of high standard and leads to a much clearer impression of how Myo7a/SANS/Harmonin and Myo7b/ANKS4B/Harmonin complexes form in microvilli. The quality of the presentation is lacking for work of such high caliber. First and foremost, in its present form, it will be exceedingly difficult for anyone outside of a few experts to follow the story presented in the manuscript. Somehow the authors need to do a better job of conveying the importance of their findings and how they differ from or enlighten the earlier studies. The figures need to be improved.

Thank you for praising the quality of our work. We appreciate the critics on our presentation and hope we have improved our figures and text according to the comments.

A few detailed comments are listed below

The first page of the introduction is excellent, but shortly thereafter it degenerates into Alphabet soup. This might be common in the field (looking back at the previous papers), but it does not make the literature accessible to the general reader.

We thank the reviewer for pointing this out, upon re-reading the Introduction it was indeed clear that the second half of this section might be difficult for a general reader to follow. The Introduction has been revised and every effort has been made to make the relevant background information and rationale for the study accessible for all readers.

Page 4 of the introduction. It would be very helpful to the non-specialist to describe up-front what structures are already available for the Myo7 tail domains. This would help place the current studies in context. There is an implication in the text of earlier structural work, but the reader has to dig out what is known. A direct statement is essential.

Agreed. A clarifying sentence has been added.

“Structures of the MF1 in complex with the CEN domain have been determined for both Myo7a-SANS and Myo7b-ANKS4B^{21,22}” (see pg 4)

Line 98. The acronym PBM should be defined here and not later (line 161)

Done, PBM is now defined upon its first use. (see pg 4)

Line 143. Are three decimal places justified for the r.m.s. deviations? The authors should read the manuscript carefully for their claims of significance and adjust accordingly.

The reviewer is correct to raise this point - we have changed the r.m.s.d. values to include only two decimal in the text and checked the text to be sure that the significance is appropriate for all of the measurements presented.

Page 5 and onwards. It would be helpful to include a description of the buried surface area for the interactions between the MyTH4-FERM domains and the Harmonin domain. This should include both the Cter and the PDZ domains (separately). This should describe both the total area and the hydrophobic and polar contributions. This would place the structures in context and would allow the general reader to gain an understanding of the nature of the interactions.

The requested information has now been added for both the Myo7a and Myo7b MF2-PDZ3c structures.

see page 6: “Binding of PDZ3c to MF2 results in the buried solvent accessible surface area of 1276 Å² (484 Å² from PDZ3 domain and 792 Å² from Cter).”

see also Supp Fig 1e legend: “Binding of PDZ3c to Myo7a MF2 results in the buried solvent accessible surface area of 1135 Å² (400 Å² from PDZ3 domain and 735 Å² from Cter).”

Line 248 and onwards. The authors need to explain the division of the CEN domain into two regions CEN1 and CEN2 otherwise it is not clear where these regions, divisions, or perhaps

domains come from. This could be defined in Figure 1B with a change to the cartoon and caption.

We have updated Fig 1B as well as its respective legend to introduce CEN1 and CEN2. We also refer to Figure 3a where the alignment of CEN region is provided. We further clarify this point in the Introduction and in the legend to Figure 1b (see page 4 and Figure 1).

“The N-terminal MF domain (MF1) binds to the CEN domain of SANS/ANKS4B, which can be further divided into CEN1 and CEN2 regions.”

Legend, Fig 1b: “The CEN domain of ANKS4B/SANS is comprised of two highly conserved CEN regions (labelled 1 and 2) connected by a short linker (see also Fig. 3a).”

Line 318. Although the interaction between NPDZ1 and PDZ3c is weak (it should be weak!) its effect will be considerably higher because of the covalent linkage between the domains which will dramatically increase the apparent concentration.

Thank you for this point - a new sentence has been added that indicates that the interactions are likely to be weaker in the full-length protein (see page 12).

“Although the detected affinity is not high, covalent linkage between these domains in FL-harmonin-a1 will drastically increase the apparent local concentration, favouring this interaction.”

Line 352 The discussion covers some very interesting territory, but it is too long (almost four pages of text). There is a lot of speculation associated with this text that could be truncated without loss (wherever the word "likely" occurs some text can be removed).

It is true that the Discussion is lengthy, there were a number of points that we wished to highlight and the topic is rather complicated. However, we agree that the Discussion could benefit from being more focused and concise.

The Discussion has been revised extensively and we have made every effort to shorten it by, in part per Reviewer-1's comment, moving some of the conclusions from the various experiments into the Results. The Discussion now has been shortened and is less than 3 pages.

Figure 2A. the electron density adds nothing but noise to this figure and should be removed. Much better density is shown in the supplementary material.

We have removed the density in Fig 2A and in the text now refers the reader to the nice density in Supplementary Fig 1B.

Figure 2B. The authors should decide what is important to show in this figure and downplay the rest. I believe it is the Cter that is important, but it is lost in the details of the MyTH4 and FERM domains. Not-fancy helices might help in Pymol. You might also consider toning down the colors and reducing the width of the ribbons.

Thank you for the suggestions. The fancy helices have been removed and we used transparency to tone down the colors. We also removed the spheres and some hydrogen

bonds that are not necessary. We hope the C-ter motif is better highlighted now.

Figure 2C. It is very hard to see the small differences in this figure with the choice of colors and widths of secondary structural elements.

We have used thinner helices and a darker grey for the apo-structure. With the small rotation of F2 and F3 lobes highlighted with arrows, we hope this is now clearer.

Figure 1D. This might belong in the supplementary material. The actual information content relative to the discussion is quite small. It is a very difficult figure to follow. The text is very small. Also the distances should not be displayed to two decimal places. The panel is very noisy and will not print well. Ideally the authors should edit the figure and remove the redundant information and only include that which is meaningful to the paper.

We believe that the reviewer is referring to Figure 2D.

Although it is still a rather busy figure, this representation illustrates all the interactions between the C-ter of harmonin-a and all three FERM lobes that a cartoon structure figure cannot. We also think this figure highlights the small differences between 7a and 7b well. We have cleaned up this figure by removing the waters and the distance labels for the hydrogen bonds. We also reorganized the labels and increased the font size to make the figure clearer.

Figure 3. There is too much information in this figure. Again the division of CEN into CEN1 and CEN2 based on sequence and structural studies should have been explained earlier. A non-expert might ask why CEN2 is restricted to only a short segment of the highly conserved region shown here. Panels B, C, E, and F are too small. The text in Panel C is completely illegible. Panel D is really a table not a figure. Panels E and F are really important since they define the model arising from this work. They are far too small. Panel F could use some more explanation (labels) within the figure to be truly useful.

Figure 3 has now been revised to try to make the information more accessible to readers.

Panel D is now new Table-2 and Panels B, C, E, F (now b, c, d, e, respectively) have been enlarged. We have also added labels in panel F (now panel-e) to help highlight the models arisen from this work.

The short length of CEN2 follows the original designation by Wu et al (2011) Science, who first showed studied the interaction of this region of SANS with the Myo7A MF1, which is the convention now adopted by those working on these adaptor proteins.

Figure 4. The interesting visual information in this figure is the location of the mutations in the interface and on the surface. This is lost in the current rendition because the colors from the ribbon dominate the visual experience. The authors should downplay the ribbons and Crayola colors and attempt to bring out the positions of mutations. The authors might consider using a level of transparency for the ribbons or muted colors.

Thank you for the suggestions. The ribbons are now thinner and we have switched to softer, more muted colors for the MF. We hope this brings out the positions of the mutations much more. A Supplemental movie S2 was also included to help the reader see where each mutation

is located.

Figure 5. There is a lot of great information in this figure, but the text is too small in Panel B and the labels in Panels C and D are also tiny in places.

Panel A has been moved into new Table 4 and enlarged the figures and labels presented in Panels C and D (now panels b and c, respectively) to make all of the information easier to see.

Figure 6 contains a lot of speculation that is beyond the scope of the current paper.

We appreciate that Figure 6 contains a lot of speculation, but it is linked to the Discussion that the reviewer also points out covers some interesting points.

We think that in order to better illustrate the potential interplay of different isoforms and the idea that Myo7a can assemble into different “complex assemblies” depending on the presence and amount of its partners (other members of the tripartite complex), it is necessary to include a model to clarify how we envision this happening.

The data presented here do provide a starting point for this speculative model and providing additional support for all aspects of the model are clearly beyond the current scope of the paper. However we think that it is important to extend the potential implications of the results from this study and that such speculative models should be addressed in future studies.

Just to be clear, this is not a model deduced solely from our current data, it takes into account the work of others in the field. We have changed the captions in the lower left and right-hand panels of Fig 6 to read “**Model** of compact transport tripartite complex” and a “**Proposed** minimal machinery for tension generation at UTLD by the tripartite complex”, respectively, to highlight that these functions are not directly addressed here. We also clarify in the legend with the title changed to “The **proposed** Myo7a Tip Link complex” and added the following description: “...the **proposed** model of this complex consists of harmonin-a, SANS, and Myo7a is shown in the blue box. Brown box: **a proposed**, much-simplified model **showing potential mechanism** of Myo7a recruitment to the UTLD based on known interactions within the tripartite complex.” We further added a sentence to clarify that our proposed models require further validation to the Discussion. (see page 14)

“Whether this simplest network or other types of assembly between harmonin and Myo7 molecules can form and would be adequate to apply force on cadherin tails requires further studies.”

Finally, we are careful to indicate that the models presented here are proposals and not yet supported by direct experimentation. We start the discussion of the models as follows (see page 13):

“A simple conceptual model for the potential role of harmonin-a in recruiting multiple Myo7a motors can be proposed.”

REVIEWERS' COMMENTS:

Reviewer #1 (Remarks to the Author):

The authors responded to all of the many concerns raised by this reviewer and in every case modified the manuscript appropriately. The text is more cohesive and scholarly. Overall the study is an outstanding contribution. Figures are also much improved. My apologies for being hypercritical, but some of the figure panels still don't have a professional appearance, from my vantage point. For example, figure 3 panel b uses a variety of font styles different from other panels in the same figure and which differ from other figures.

On line 96 of the revised manuscript, please delete "highly". A sequence is or is not homologous to some other sequence. There is no degree of homology. There is degree of similarity or degree of identity. Evolutionary biologist recoil at degrees of homology.

Point-by-point response to reviewer's comments:

Reviewer #1 (Remarks to the Author):

The authors responded to all of the many concerns raised by this reviewer and in every case modified the manuscript appropriately. The text is more cohesive and scholarly. Overall the study is an outstanding contribution. Figures are also much improved. My apologies for being hypercritical, but some of the figure panels still don't have a professional appearance, from my vantage point. For example, figure 3 panel b uses a variety of font styles different from other panels in the same figure and which differ from other figures.

We thank the reviewer for recognizing the work and the improvement on the manuscript. We have carefully updated the figures to make all fonts and styles as consistent as possible throughout and cleaned up the figures wherever possible. Figure 3b - font is consistent and we changed the labeling color to black to have a cleaner look. Figure 3c - the labeling and font size of all MST curves are now consistent as in figure 4a and figure 5a. Figure 3d - the font and sizes of the labeling are consistent with that in 3b, while the color is coordinated with the color used for each structure domain. Figure 4b and 4c - the labeling of residues are now in the same font and size. Font style in figure 1a and figure 6 are also changed to be consistent with the rest of the figures.

On line 96 of the revised manuscript, please delete "highly". A sequence is or is not homologous to some other sequence. There is no degree of homology. There is degree of similarity or degree of identity. Evolutionary biologist recoil at degrees of homology.

We removed "highly" in this sentence.